

# Adaptations to cursoriality and digit reduction in the forelimb of the African wild dog (*Lycaon pictus*)

Heather F. Smith[1,2,3,4], Brent Adrian[1], Rahul Koshy[3], Ryan Alwiel[4] and Aryeh Grossman[1,3,4]

[1] Department of Anatomy, College of Graduate Studies, Midwestern University, Glendale, AZ, USA
[2] School of Human Evolution and Social Change, Arizona State University, Tempe, AZ, USA
[3] Arizona College of Osteopathic Medicine, Midwestern University, Glendale, AZ, USA
[4] College of Veterinary Medicine, Midwestern University, Glendale, AZ, USA

## ABSTRACT

**Background:** The African wild dog (*Lycaon pictus*), an endangered canid native to southern and eastern Africa, is distinct among canids in being described as entirely tetradactyl and in its nomadic lifestyle and use of exhaustive predation to capture its prey instead of speed, strength, or stealth. These behavioral and morphological traits suggest a potentially unique set of adaptations.

**Methods:** Here, we dissected the forelimbs of an adult male *L. pictus* specimen and performed detailed descriptions and quantitative analyses of the musculoskeletal anatomy.

**Results:** Statistical comparisons of muscle masses and volumes revealed that *L. pictus* has relatively smaller wrist rotators (mm. pronator teres, pronator quadratus, supinator) than any other included carnivoran taxon, suggesting adaptive pressures for antebrachial stability over rotatory movement in the carpus of *L. pictus*. While a complete digit I is absent in *L. pictus*, a vestigial first metacarpal was discovered, resulting in changes to insertions of mm. extensor digiti I et II, abductor (et opponens) digiti I and flexor digiti I brevis. Mm. anconeus, brachialis and flexor carpi ulnaris caput ulnare all have more extensive origins in *L. pictus* than other canids suggesting an emphasis on posture and elbow stability. M. triceps brachii caput laterale has a larger origin in *L. pictus* and m. triceps brachii caput longum has an additional accessory head. Electromyographic studies have shown this muscle is active during the stance phase of trotting and galloping and is important for storing elastic energy during locomotion. We interpret these differences in size and attachments of muscles in *L. pictus* as adaptations for long distance running in this highly cursorial species, likely important for exhaustive predation. Absence of a full digit I in *L. pictus* may increase speed and stride length; however, the retention of a vestigial digit permits the attachment of reduced pollical muscles which may provide additional stability and proprioception to the carpus.

Corresponding author
Heather F. Smith,
Heather.F.Smith@asu.edu

## INTRODUCTION

The African wild dog (*Lycaon pictus*), also known as the African painted dog or Cape hunting dog, is unique among canids in many ways. Phylogenetically, it falls as the outgroup of the "wolf" clade of canids (*Toh et al., 2005*). However, unlike other canid species, *L. pictus* utilizes exhaustive predation instead of speed, strength, or stealth to hunt and capture its prey. Sophisticated hunting behaviors are deployed, and some packs exhibit quorums on decisions to hunt (*Walker et al., 2017*). They communicate their vote via "sneezing" or a sharp exhale through their nostrils (*Walker et al., 2017*). They feed primarily on antelope, which they hunt by running the prey to exhaustion. Larger prey are hunted in packs, while smaller prey such as rodents and hares are hunted by individual dogs (*Woodroffe et al., 2007*). *L. pictus* achieves higher successful hunt rates (greater than 60%) than even lions (27–30%) and hyenas (25–30%), although they frequently lose their kills to these larger predators (*Creel & Marusha, 1998*). It is a hypercarnivore, a dietary adaptation which results in specialized craniodental morphology compared to other canids. *L. pictus* also exhibits a nomadic lifestyle with packs traveling up to 50 km per day (*Woodroffe, McNutt & Mills, 2004*) and geographically extensive home ranges of between 560 and 3,000 km$^2$ (*Castello, 2018*). These unique hunting and behavioral characteristics suggest that the anatomy of *L. pictus* should be adapted for long-distance, endurance running. In particular, their forelimb anatomy likely exhibits adaptations to compensate for this type of lifestyle. However, this hypothesis has not been tested.

An additional morphological trait distinguishing *L. pictus* from other caniforms is its reported absence of a manual digit I (pollex) or "dewclaw". The absence of digit I may allow for increased speed and stride length in *L. pictus*, thus facilitating long-distance pursuit of prey (*Van Valkenburgh, 1987*; *Creel & Creel, 2002*). Across mammals, there is a trend among cursorial predators towards digit loss and limb elongation, both of which enable speed and enhance prey capture capabilities (*Van Valkenburgh, 1987*). Although the fossil record of *Lycaon* is poorly known, limited material suggests that *Lycaon, Canis* and *Cuon* may have diverged from a common ancestor in the Pliocene (*Kurtén, 1968*; *Wayne, 1993*, *Stiner et al., 2001*). The subsequent reduction of the first digit in the *Lycaon* lineage evolved shortly thereafter, alongside dental adaptations for hypercarnivory (*Martínez-Navarro & Rook, 2003*; *Hartstone-Rose et al., 2010*). A recent study on *L. pictus* genomics identified several genes associated with digit I loss (*Chavez et al., 2019*). Digit reduction in *L. pictus* develops through apoptosis of the first digit during embryonic development, specifically via a pathway that generally regulates apoptosis of interdigital tissue (*Chavez et al., 2019*). Nevertheless, it is currently unknown how the lack of digit I in *L. pictus* may affect the morphology and attachment points of the numerous muscles of the antebrachium and manus.

In addition to its unique anatomical and behavioral adaptations, *L. pictus* is also an important study subject due to its conservation status. It is classified as Endangered by the International Union for Conservation of Nature (IUCN) with as few as 1,400 mature individuals present in the wild (*Woodroffe & Sillero-Zubiri, 2020*). Much of the population inhabits savanna and arid zones in southern and southeastern Africa. The population size

is declining, primarily due to human-caused habitat fragmentation (*Woodroffe & Sillero-Zubiri, 2020*), and there is currently no detailed documentation of their anatomy. The descriptions, illustrations, and comparative quantitative analyses of their forelimb morphology provided here can serve as a useful reference for exotic animal veterinarians and other clinicians, as well as conservation scientists who work with these animals.

# MATERIALS AND METHODS

## Dissection and muscle descriptions

This research was conducted on the left and right forelimbs of an adult male African wild dog (*L. pictus*). The captive specimen was donated to the Arizona Research Collection for Integrative Vertebrate Education and Study (ARCIVES) at Midwestern University by the Arizona Center for Nature Conservation/Phoenix Zoo. The specimen was received by ARCIVES post-necropsy, and consequently the limbs had been disconnected from the trunk and several extrinsic back muscles had been disconnected. Following necropsy, the forelimbs were preserved by submersion in a 10% formaldehyde solution. The body condition of the animal, despite being disarticulated, was assessed to be good and muscle tone was moderate.

Dissections took place in the Department of Anatomy, College of Veterinary Medicine at Midwestern University (Glendale, AZ, USA). Photodocumentation for all structures was performed using a Nikon DSLR camera throughout the dissection process. Using a veterinary atlas and published descriptions as reference, exposed muscles and neurovasculature were compared to other caniforms for identification and comparison purposes. In particular, we utilized descriptive studies on the myology of the domestic dog, *Canis familiaris*, (*Evans & De Lahunta, 2013*), pampas fox, *Lycalopex gymnocercus* (*De Souza Junior et al., 2018*) red panda, *Ailurus fulgens* (*Fisher et al., 2009*), giant panda, *Ailuropoda melanoleuca* (*Davis, 1964*), and lesser grison, *Galictis cuja* (*Ercoli et al., 2015*) as primary comparative references. Each muscle was described in detail, including its gross morphology, fiber direction, number of bellies, attachment points, and relationships to other structures. As the dissection progressed, muscles were carefully detached from the forelimb bones to identify specific osteological attachment sites. We mapped the origins and insertions of muscles by drawing onto transparent sheet protectors over printed photographs of comparative bones. These maps were then converted into digital illustrations to clarify muscle descriptions. We also collected quantitative muscle data: Mass was recorded with an Accuteck® electronic balance, and lengths and widths were assessed using Mitutoyo digital calipers. Ultimately, morphology was compared to that of other published carnivoran species to identify adaptations for cursoriality, long-distance endurance running, and digit reduction in *L. pictus*.

## Quantitative analyses

To assess how forelimb muscular proportions of *L. pictus* compare to those of other carnivorans, we conducted quantitative statistical analyses of muscle data, primarily following the approach of *Taverne et al. (2018)*. First, the mass of each muscle (Table 1) was used to calculate the volume using the known muscle density value of 1.06 g cm$^{-3}$

**Table 1 Quantitative data on forelimb muscles of the adult male *Lycaon pictus* specimen dissected in the present study.**

| Muscle | Mass (g) | Proportion (%) |
|---|---|---|
| M. acromiodeltoideus | 19.0 | 0.0233 |
| M. spinodeltoideus | 23.0 | 0.0282 |
| M. supraspinatus | 124.0 | 0.1522 |
| M. infraspinatus | 95.0 | 0.1166 |
| M. subscapularis | 75.0 | 0.0921 |
| M. teres major | 34.0 | 0.0417 |
| M. teres minor | 5.0 | 0.0061 |
| M. biceps brachii | 32.0 | 0.0393 |
| M. brachialis | 18.0 | 0.0221 |
| M. triceps brachii | 240.0 | 0.2947 |
| M. anconeus | 5.0 | 0.0061 |
| M. brachioradialis | 10.0 | 0.0123 |
| M. extensor carpi ulnaris | 13.0 | 0.0160 |
| M. extensor digitorum lateralis | 5.0 | 0.0061 |
| M. extensor digitorum communis | 10.0 | 0.0123 |
| M. extensor digiti I and II | 0.5 | 0.0006 |
| M. abductor digiti I longus | 1.0 | 0.0012 |
| M. extensor carpi radialis—longus + brevis | 27.0 | 0.0331 |
| M. supinator | 0.5 | 0.0006 |
| M. pronator teres | 1.0 | 0.0012 |
| M. palmaris longus | 16.0 | 0.0196 |
| M. flexor carpi ulnaris, ulnar head | 6.0 | 0.0074 |
| M. flexor carpi ulnaris, humeral head | 14.0 | 0.0172 |
| M. flexor carpi radialis | 6.0 | 0.0074 |
| M. flexor digitorum profundus | 43.0 | 0.0528 |
| M. pronator quadratus | 0.5 | 0.0006 |
| Total mass of included muscles | 814.5 | 1.0000 |

(*Mendez et al., 1960*). Volumes were then $\log_{10}$ transformed to normalize the data and regressed against the combined weight of the forelimb muscles as this accounts for differences in overall body size in comparisons to other carnivoran taxa.

We combined muscles into functional groups as defined by *Taverne et al. (2018)*: elbow extensors, elbow flexors, wrist extensors, wrist flexors, and wrist rotators. The additional categories of humeral and scapular muscle groups in *Taverne et al. (2018)* could not be applied to the current dataset due to the damage of certain key muscles (e.g., mm. trapezius, rhomboideus, latissimus dorsi) in our *L. pictus* specimen during necropsy. We compiled comparative data on forelimb muscle volume proportions for 17 felid and canid species from *Taverne et al. (2018)* and compared those to *L. pictus*. We conducted a Principal Component Analysis using the aforementioned volume proportions for the forelimb muscle groups using SPSS 25 (IBM Corp., Armonk, NY, USA).

We also compiled comparative data on individual muscle masses from published literature, including *Cuon alpinus* and *Vulpes vulpes* (*Taverne et al., 2018*), *Galictis cuja* (*Ercoli et al., 2015*), *Lynx lynx* (*Viranta et al., 2016*) and *Leopardus pardalis* (*Julik et al., 2012*). Relative proportions of muscles were calculated as described above and compared graphically and via Principal Component Analysis in SPSS 25 (IBM Corp., Armonk, NY, USA). This step permits the evaluation of separate muscles within each functional group, as well as the inclusion of additional scapular muscles not included in the functional group analyses.

While it has been documented that specimens preserved in formalin may experience shrinkage (*Fox et al., 1985*), this phenomenon is unlikely to have a significant impact on the analyses conducted here for several reasons. First, as the data compared here are percentages, it is likely that all muscles are impacted equally, and thus the ratios should be accurate. Second, in the comparative sample, the muscle data of *Taverne et al. (2018)* also used formalin-fixed specimens, so the values are directly comparable.

# RESULTS

## Vestigial digit 1

While the majority of the bony anatomy of *L. pictus* generally coincides with standard canid osteological patterns, there is one notable exception. During reflection of the fascia around the medial carpus, a small bony protrusion emerged. Extensive cleaning revealed an unexpected vestigial metacarpal I (Fig. S1). The bone is slender, tapering in the center and rounded on both ends, resembling an elongated dumbbell. It is 19.8 mm long, making it approximately 30% the length of metacarpal II (65 mm long). It measures 4.5 mm wide at the base, 2.9 mm wide at the midshaft, and 4.7 wide at its head. No accompanying sesamoid bone was observed, and there were no associated phalanges. The metacarpal I serves as a muscle attachment site for several typical carnivoran pollical muscles, discussed in further detail below.

## Quantitative analyses

In the Principal Component Analysis, the first three PCs explain greater than 90% of the variance: PC1 51.3%, PC2 30.0% and PC3 10.9%. Terrestrial and arboreal species separated along PC1 axis. The elbow extensor group loaded most negatively along this axis (−0.989), while wrist rotator group loaded most positively (0.854). Since these muscle groups generated the greatest degree of separation among locomotor groups, we performed a follow-up Ordinary Least Squares regression analysis regressing wrist rotator values over elbow extensor values. The resulting plot illustrates an almost-perfect separation between arboreal and terrestrial taxa (Fig 1; Table 2). Additionally, larger-bodied terrestrial species that spend more time running cluster together, separate from the smaller-bodied short-distance scampering taxa (Fig 1). *L. pictus* clusters with the runners, along with *Vulpes vulpes* (red fox), *Cuon alpinus* (dhole), *Acinonyx jubatus* (cheetah) and *Hyaena hyaena* (striped hyena). In particular, these species are characterized by relatively smaller proportions of wrist rotators and relatively larger elbow extensors. Of all included taxa, *L. pictus* has the smallest wrist rotator group.

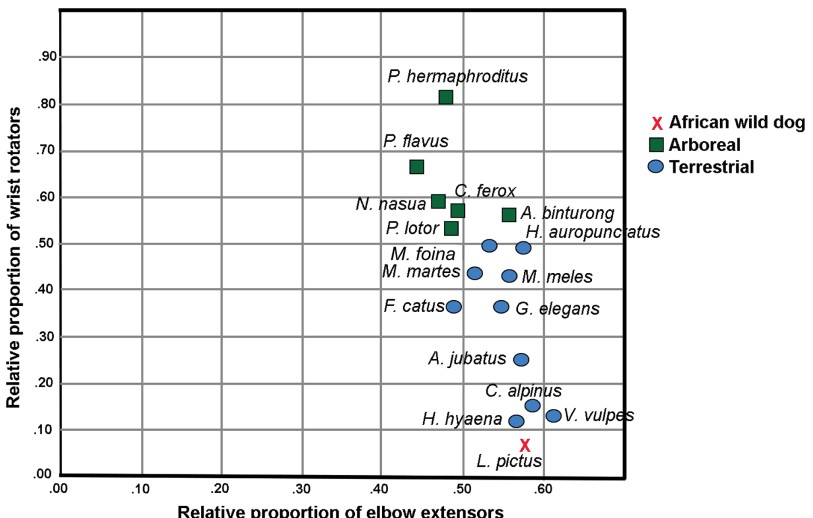

**Figure 1 Regression plot of relative proportions of wrist rotator functional group to elbow extensor functional group in *L. pictus* and the comparative carnivoran sample (*Taverne et al., 2018*).** Note the position of *L. pictus* which has the smallest wrist rotator group in the sample.

Of the taxa for which individual muscle mass data were available, *L. pictus* was found to possess smaller proportions of mm. pronator teres, pronator quadratus, supinator, abductor digiti I longus, extensor digiti I et II than any other included taxon (Table 3). *L. pictus* has the highest values in the sample of m. acromiodeltoideus (also high m. spinodeltoideus) and m. infraspinatus. The teres minor of *L. pictus* is also relatively large, a trait shared with *L. pardalis*. Compared to the felids in the sample, all three canids shared larger mm. articularis humeri and triceps brachii, and smaller mm. teres major, biceps brachii, abductor digiti I longus, supinator, pronator teres and pronator quadratus.

Compared to the other canids in the sample, *Cuon alpinus* and *Vulpes vulpes*, *L. pictus* has dramatically proportionally larger mm. deltoideus, infraspinatus, brachialis, flexor carpi ulnaris caput humerale, and flexor carpi radialis. It has relatively smaller mm. triceps brachii (all heads combined), anconeus, extensor digitorum communis, abductor digiti I longus, supinator, pronator teres, pronator quadratus, flexor digitorum superficialis, and flexor carpi ulnaris caput ulnare. The two larger-bodied canids (>15 kg), *C. alpinus* and *L. pictus*, share larger mm. supraspinatus, extensor carpi ulnaris, extensor digitorum lateralis, and smaller m. extensor digiti I et II.

## Muscle descriptions

### Extrinsic muscles of the shoulder

*M. trapezius*

This muscle was extensively damaged during necropsy, so its origin and the morphology of its proximal fibers could not be evaluated. Its distal fibers consist of a thin muscle belly with muscle fibers coursing inferolaterally towards the spine of the scapula (Fig. 2). The muscle lies superficial to the supraspinatus (Fig. 2). Near its insertion, it courses adjacent to the m. omotransversarius, and their bellies adhere to each other by a thin band of fascia.

**Table 2 Volumes and relative proportions of functional forelimb muscle groups in *Lycaon pictus* and comparative carnivoran taxa (*Taverne et al., 2018*).**

| Taxon | Sum | Elbow extensors | Elbow flexors | Wrist extensors | Wrist flexors | Wrist rotators | Elbow extensors-proportion | Elbow flexors-proportion | Wrist extensors-proportion | Wrist flexors-proportion | Wrist rotators-proportion |
|---|---|---|---|---|---|---|---|---|---|---|---|
| *Lycaon pictus* | 554.72 | 318.87 | 99.06 | 52.83 | 80.19 | 3.77 | **0.575** | 0.179 | 0.095 | 0.145 | **0.007** |
| *Mustela putorius* | 24.62 | 12.68 | 3.86 | 2.46 | 4.53 | 1.09 | 0.515 | 0.157 | 0.100 | 0.184 | 0.044 |
| *Vulpes vulpes* | 144.07 | 87.70 | 22.82 | 12.42 | 19.06 | 2.07 | 0.609 | 0.158 | 0.086 | 0.132 | 0.014 |
| *Herpestes auropunctatus* | 5.25 | 2.89 | 0.79 | 0.52 | 0.79 | 0.26 | 0.550 | 0.150 | 0.099 | 0.150 | 0.050 |
| *Procyon lotor* | 34.52 | 16.93 | 6.68 | 3.40 | 5.67 | 1.84 | 0.490 | 0.194 | 0.098 | 0.164 | 0.053 |
| *Acinonyx jubatus* | 1060.2 | 606.00 | 209.00 | 83.00 | 136.20 | 26.00 | 0.572 | 0.197 | 0.078 | 0.128 | 0.025 |
| *Martes martes* | 41.86 | 21.58 | 7.14 | 4.16 | 7.13 | 1.85 | 0.516 | 0.171 | 0.099 | 0.170 | 0.044 |
| *Martes foina* | 31.31 | 16.00 | 5.14 | 3.25 | 5.35 | 1.57 | 0.511 | 0.164 | 0.104 | 0.171 | 0.050 |
| *Meles meles* | 250.04 | 134.94 | 33.12 | 22.24 | 48.81 | 10.93 | 0.540 | 0.132 | 0.089 | 0.195 | 0.044 |
| *Galidia elegans* | 12.03 | 6.52 | 1.99 | 1.12 | 1.96 | 0.44 | 0.542 | 0.165 | 0.093 | 0.163 | 0.037 |
| *Cryptoprocta ferox* | 64.06 | 31.74 | 11.18 | 6.83 | 10.62 | 3.69 | 0.495 | 0.175 | 0.107 | 0.166 | 0.058 |
| *Paradoxurus hermaphroditus* | 34.97 | 16.47 | 6.80 | 3.84 | 5.04 | 2.82 | 0.471 | 0.194 | 0.110 | 0.144 | 0.081 |
| *Potos flavus* | 42.09 | 18.52 | 8.21 | 4.52 | 8.07 | 2.77 | 0.440 | 0.195 | 0.107 | 0.192 | 0.066 |
| *Cuon alpinus* | 496.92 | 293.00 | 76.00 | 42.49 | 78.00 | 7.43 | 0.590 | 0.153 | 0.086 | 0.157 | 0.015 |
| *Hyaena hyaena* | 1162 | 661.00 | 201.00 | 105.90 | 181.00 | 13.10 | 0.569 | 0.173 | 0.091 | 0.156 | 0.011 |
| *Nasua nasua* | 54.97 | 25.43 | 9.03 | 4.41 | 12.83 | 3.27 | 0.463 | 0.164 | 0.080 | 0.233 | 0.059 |
| *Felis silvestris catus* | 42.4 | 20.75 | 8.44 | 5.39 | 6.27 | 1.55 | 0.489 | 0.199 | 0.127 | 0.148 | 0.037 |
| *Arctictis binturong* | 434 | 237.40 | 72.90 | 35.20 | 63.90 | 24.60 | 0.547 | 0.168 | 0.081 | 0.147 | 0.057 |

**Note:**
*Lycaon pictus* has the lowest proportion of wrist rotators and among the largest elbow extensors in the sample (values indicated in bold).

It inserts onto the proximal two-thirds of the cranial aspect of the scapular spine via fleshy fibers (Figs. 2 and 3A). A separable pars thoracica and pars cervicalis could not be identified in this specimen, likely due to damage during necropsy. M. trapezius acts to move the scapula cranially and dorsally, thus elevating the forelimb, as well as stabilize the scapula.

### M. latissimus dorsi

The origin of this muscle was damaged during necropsy. It is a thick sheet of muscle, slightly thicker than m. teres major (Fig. 4). Remaining muscle fibers run perpendicular to the length of the humerus (Fig. 4). There is an aponeurosis between m. latissimus dorsi and m. teres major, and the muscles fuse distally to insert on the teres major tuberosity of the humerus via a thick conjoined tendon (Figs. 3A and 5). M. latissimus dorsi acts to retract and abduct the forelimb. When the forelimb is fixed, it moves the trunk cranially.

### M. omotransversarius

The origin of this muscle was damaged during necropsy. It is a thin muscle belly similar in appearance to the m. trapezius but narrower (Fig. 2). Muscle fibers course caudolaterally,

**Table 3 Relative proportions of individual forelimb muscles in *L. pictus* and comparative sample.**

| Muscle | Lycaon pictus Caniformia | Cuon alpinus Caniformia | Vulpes vulpes Caniformia | Galictis cuja Caniformia | Leopardus pardalis Feliformia | Lynx lynx Feliformia |
|---|---|---|---|---|---|---|
| M. acromiodeltoideus | 0.0233 | 0.0191 | 0.0147 | 0.0193 | 0.0199 | 0.0195 |
| M. spinodeltoideus | 0.0282 | 0.0143 | 0.0273 | 0.0126 | 0.0244 | 0.0211 |
| M. supraspinatus | 0.1522 | 0.1561 | 0.1161 | 0.1184 | 0.1243 | 0.1069 |
| M. infraspinatus | 0.1166 | 0.0860 | 0.0883 | 0.0525 | 0.0990 | 0.0742 |
| M. subscapularis | 0.0921 | 0.0780 | 0.0899 | 0.0744 | 0.1457 | 0.0838 |
| M. teres major | 0.0417 | 0.0350 | 0.0484 | 0.0172 | 0.0664 | 0.0639 |
| M. teres minor | 0.0061 | 0.0023 | 0.0054 | 0.0009 | 0.0066 | 0.0048 |
| M. biceps brachii | 0.0393 | 0.0398 | 0.0390 | 0.0368 | 0.0635 | 0.0643 |
| M. brachialis | 0.0221 | 0.0191 | 0.0208 | 0.0340 | 0.0238 | 0.0278 |
| M. triceps brachii combined | 0.2947 | 0.3328 | 0.3586 | 0.3392 | 0.1935 | 0.2581 |
| M. anconeus | 0.0061 | 0.0096 | 0.0074 | 0.0081 | 0.0177 | 0.0084 |
| M. brachioradialis | 0.0123 | 0.0000 | 0.0000 | 0.0172 | 0.0085 | 0.0033 |
| M. extensor carpi ulnaris | 0.0160 | 0.0159 | 0.0117 | 0.0221 | 0.0181 | 0.0156 |
| M. extensor digitorum lateralis | 0.0061 | 0.0064 | 0.0049 | 0.0095 | 0.0064 | 0.0083 |
| M. extensor digitorum communis | 0.0123 | 0.0127 | 0.0127 | 0.0134 | 0.0100 | 0.0196 |
| M. extensor digiti I and II | 0.0006 | 0.0008 | 0.0012 | 0.0032 | 0.0021 | 0.0000 |
| M. abductor digiti I longus | 0.0012 | 0.0048 | 0.0064 | 0.0124 | 0.0125 | 0.0122 |
| M. extensor carpi radialis longus + brevis | 0.0331 | 0.0271 | 0.0342 | 0.0321 | 0.0256 | 0.0358 |
| M. supinator | 0.0006 | 0.0021 | 0.0020 | 0.0074 | 0.0082 | 0.0054 |
| M. pronator teres | 0.0012 | 0.0048 | 0.0065 | 0.0197 | 0.0199 | 0.0163 |
| M. palmaris longus | 0.0196 | 0.0255 | 0.0215 | 0.0266 | 0.0247 | 0.0228 |
| M. flexor carpi ulnaris, ulnar head | 0.0074 | 0.0239 | 0.0045 | 0.0254 | 0.0080 | 0.0057 |
| M. flexor carpi ulnaris, humeral head | 0.0172 | 0.0064 | 0.0126 | 0.0146 | 0.0108 | 0.0159 |
| M. flexor carpi radialis | 0.0074 | 0.0064 | 0.0059 | 0.0077 | 0.0077 | 0.0133 |
| M. flexor digitorum profundus | 0.0528 | 0.0621 | 0.0518 | 0.0716 | 0.0502 | 0.0643 |
| M. pronator quadratus | 0.0006 | 0.0050 | 0.0036 | 0.0029 | 0.0059 | 0.0045 |
| Total mass of included muscles | 814.50 | 627.97 | 172.70 | 97.58 | 258.83 | 626.09 |

**Note:**
Comparative data were taken from *Julik et al. (2012)* (*Leopardus pardalis*), *Ercoli et al. (2015)* (*Galictis cuja*), *Viranta et al. (2016)* (*Lynx lynx*) and *Taverne et al. (2018)* (*Cuon alpinus* and *Vulpes vulpes*).

perpendicular to the spine of the scapula (Fig. 2). The muscle lies superficial to the craniodistal edge of m. supraspinatus and is proximal to the origin of m. deltoideus pars acromialis (Fig. 2). It inserts onto the distal one-third of the scapular spine adjacent to the insertion of m. trapezius, and onto the proximal acromion just proximal to the insertion of m. deltoideus pars acromialis (Figs. 2 and 3A). M. omotransversarius pulls the scapula cranially, advancing the forelimb. When the forelimb is fixed, it flexes the neck.

*M. cleidobrachialis*
The origin of this muscle was damaged during necropsy. It consists of a long round muscle belly on the cranial portion of the brachium which tapers towards its insertion (Figs. 2 and 4). Muscle fibers run parallel to the long axis of the muscle (Figs. 1 and 3). The muscle

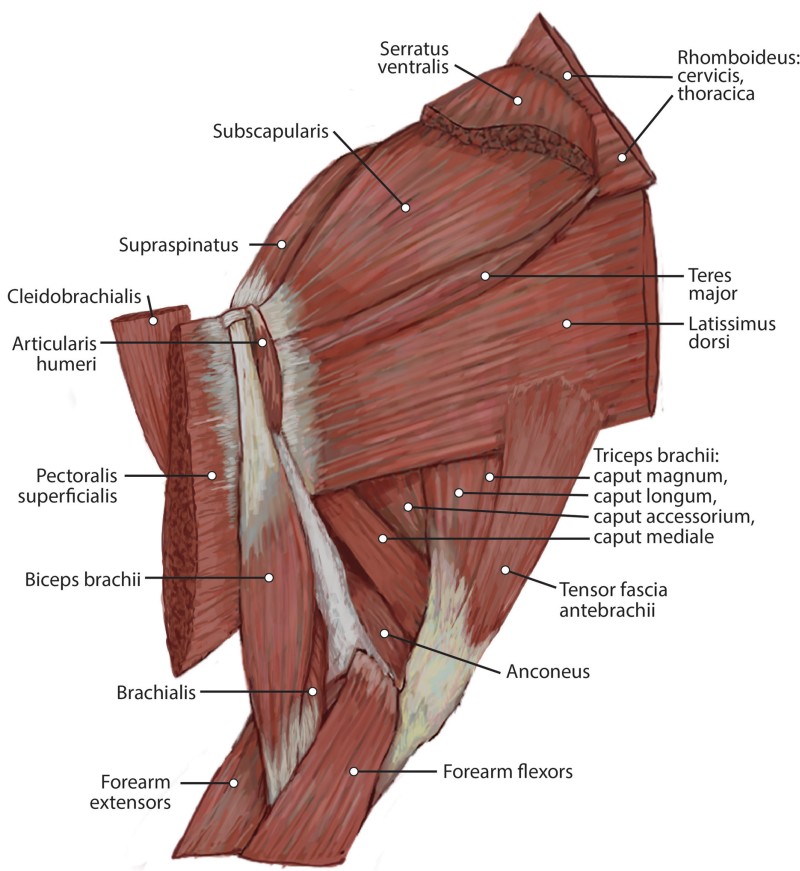

**Figure 2 Medial view of the right shoulder and brachium in *Lycaon pictus*.**

travels cranial to m. deltoideus pars acromialis and lateral to m. pectoralis superficialis (Figs. 2 and 4). M. cleidobrachialis inserts onto a small area of the craniodistal humerus between m. biceps brachii and m. brachialis muscle bellies (Figs. 2, 4 and 5A). M. cleidobrachialis draws the forelimb cranially, or flexes the neck if the forelimb is fixed.

### M. pectoralis superficialis

This muscle was extensively damaged during necropsy; however, its attachment to the humerus was preserved. Muscle fibers are positioned medial to m. cleidobrachialis and lateral to m. biceps brachii (Figs. 2 and 4). Muscle fibers course perpendicular to the long axis of the humerus (Figs. 2 and 4). It inserts along the proximal two-thirds of the cranial surface of the humerus (Figs. 2, 4, 5A and 5B). M. pectoralis superficialis adducts the forelimb and may draw the forelimb cranially or caudally depending on the portion of the muscle that is active and the position of the limb.

### M. deltoideus

Pars scapularis: This muscle originates from the scapular spine indirectly via a connection to the aponeurosis of m. infraspinatus (Fig. 2). It is a flat fusiform muscle which lies superficial to the inferior portion of the infraspinatus and completely overlies m. teres minor

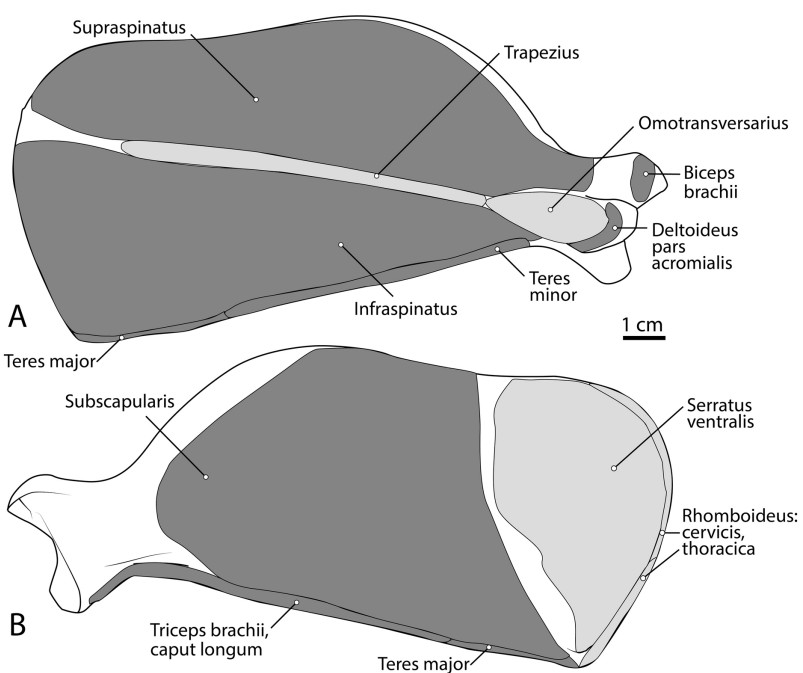

**Figure 3 Scapula muscle maps for *L. pictus* (right side): (A) lateral view, (B) medial view.** Here and in subsequent figures, dark grey indicates muscle origins and light grey indicates muscle insertions.

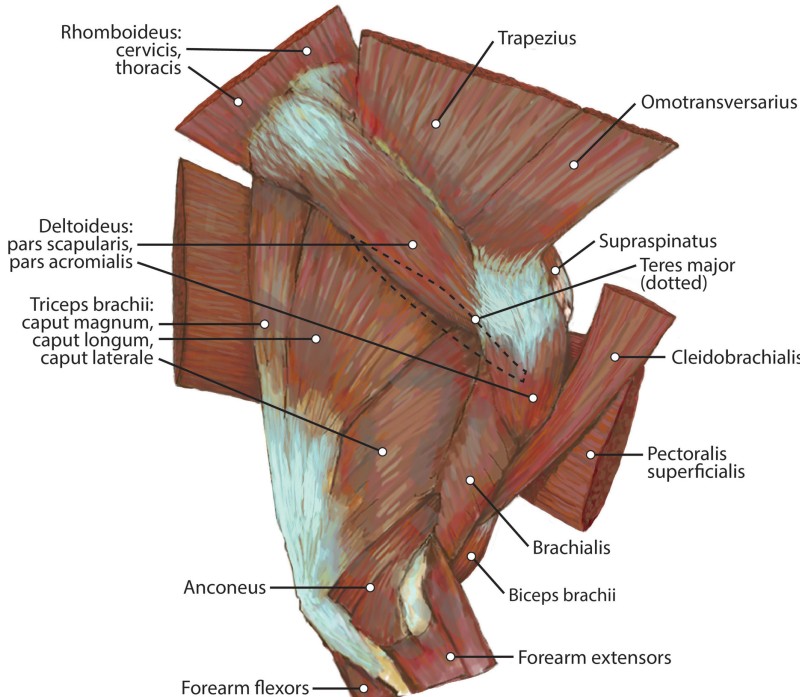

**Figure 4 Lateral view of the right shoulder and brachium in *L. pictus*.**

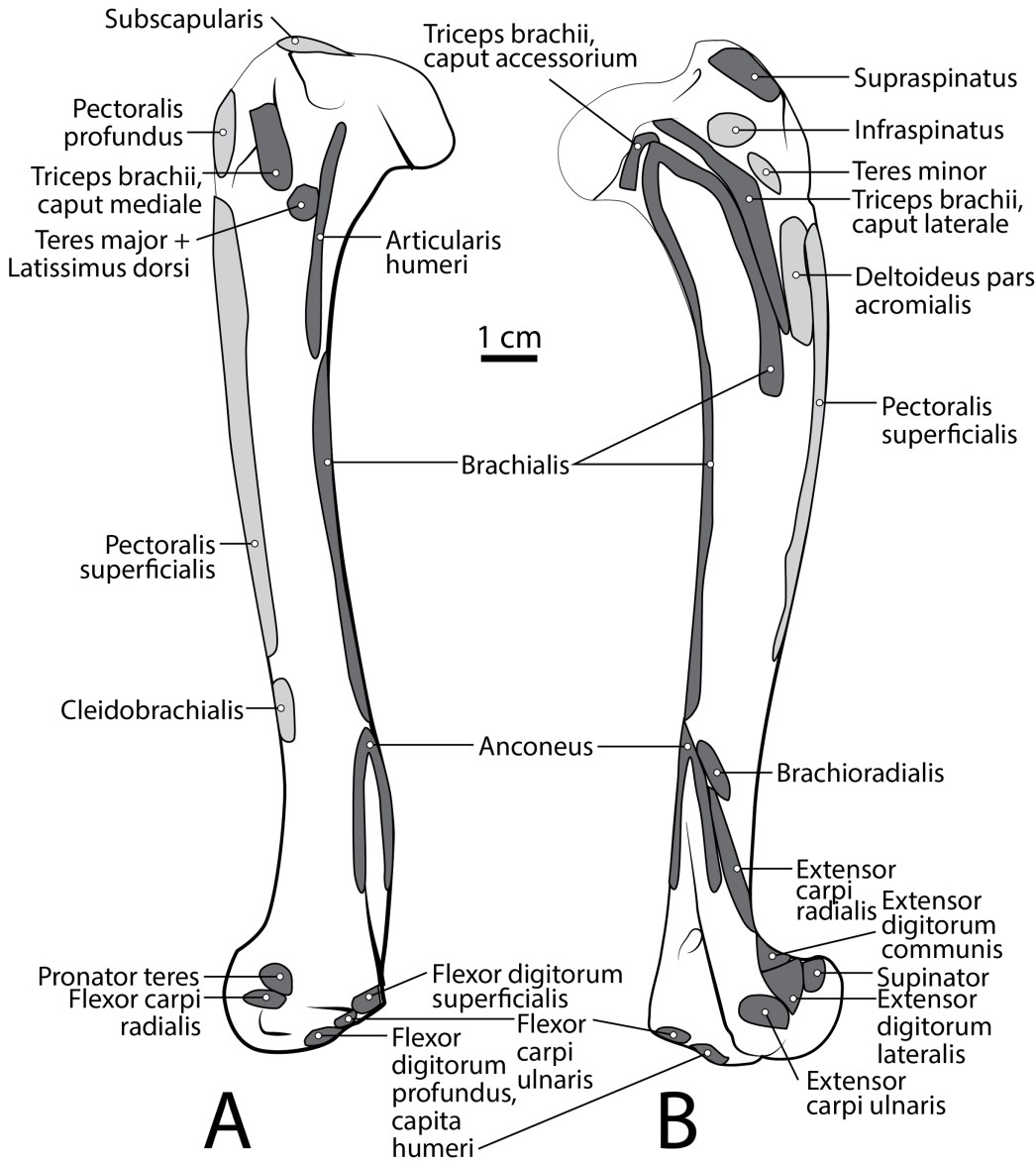

**Figure 5** Humerus muscle maps for *L. pictus* (right side): (A) medial view, (B) lateral view.

(Fig. 1). The muscle fibers run parallel to the length of the spine of the scapula (Fig. 2).
It inserts onto the m. deltoideus pars acromialis and has no direct connection to the humerus
(Fig. 2).

Pars acromialis: The origin of this muscle is from the acromion process of the scapula
distal to the insertion of m. omotransversarius (Figs. 2 and 3A). Its muscle belly is fusiform
and has a prominent aponeurosis covering it proximally (Fig. 2). The proximal muscle
fibers course parallel to the scapular spine, and then slowly curve distally to become
parallel to the humerus (Fig. 2). It has a large insertion onto the deltoid tuberosity on the
lateral humerus (Figs. 2 and 5B).

M. deltoideus flexes the glenohumeral joint and abducts the humerus.
*M. supraspinatus*

This muscle has a large, thick muscle belly originating from and fully filling the supraspinous fossa to the tip of the spine of the scapula (Figs. 2 and 3A). The muscle is elliptical in shape, similar to, but wider than m. infraspinatus. Muscle fibers run obliquely from the spine of the scapula to the distal margins of the muscle belly. An aponeurosis covers the proximal one-third of the muscle surface. There is a thick tendon in the middle of the muscle which travels two-thirds proximally up the muscle from the glenohumeral joint to the proximal point and dissipates, similar to the pattern in m. infraspinatus. It inserts onto the greater tubercle of the humerus along its cranial aspect, proximal to the insertion of m. infraspinatus (Figs. 2, 3 and 5B). As with the other muscles of the rotator cuff, m. supraspinatus stabilizes the shoulder. It also extends the glenohumeral joint, resulting in advancement of the forelimb.

*M. infraspinatus*

This muscle originates from and occupies the infraspinous fossa (Fig. 3A). It has a thick muscle belly with an elliptical shape that tapers on its proximal and distal ends. Distal portions of the muscle are deep to the m. deltoideus pars scapularis. The muscle has various muscle fiber directions with the cranial and caudal fibers coursing diagonally to converge on the midline of the muscle. The tendon adhering to the humerus is located within the muscle belly and travels three-quarters of the length of m. infraspinatus before widening and inserting on the greater tubercle of the humerus (Fig. 5B). The flat portion of the tendon parallels the spine of the scapula. M. infraspinatus has an extensive aponeurosis on its cranial portion covering about one-third of the superficial muscle. It inserts onto the greater tubercle of the humerus inferior to the insertion of m. supraspinatus (Fig. 5B). M. infraspinatus acts to laterally rotate the humerus and stabilize the glenohumeral joint. It can also contribute to flexion or extension of the shoulder, depending on the position of the humeral head relative to the glenoid when the muscle contracts.

*M. teres minor*

M. teres minor originates from the distal three-quarters of the infraspinous fossa margin (Figs. 2 and 3A). Proximally, the muscle is thin and broad, and then it thickens quickly to a spindle shape and tapers distally (Figs. 2 and 5B). Its insertion is the teres minor tuberosity on the proximolateral aspect of the greater tubercle of the humerus (Figs. 2 and 5B). M. teres minor flexes the glenohumeral joint.

*M. teres major*

The origin of this muscle was damaged during necropsy. The muscle forms a thick sheet which courses adjacent to m. latissimus dorsi and adheres to it near the origin of m. tensor fascia antebrachii on m. latissimus dorsi (Fig. 3). It inserts broadly onto the teres major tuberosity (Figs. 3 and 5A) cranial to the insertion of m. triceps brachii caput mediale. The attachment of m. teres major onto the humerus is thicker and stronger than the attachments of the rotator cuff muscles. M. teres major flexes the glenohumeral joint, drawing the humerus caudally. It may also medially rotate the humerus, resisting lateral rotation.

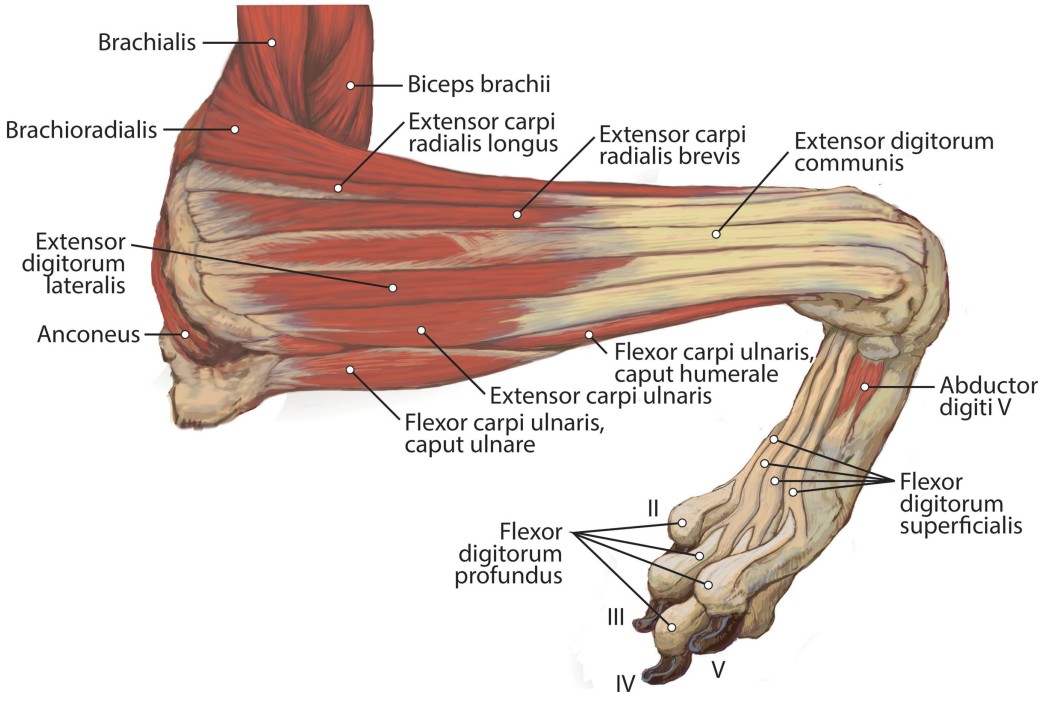

**Figure 6 Lateral view of the superficial right antebrachium in *L. pictus*, including mm. biceps brachii, brachialis, and anconeus.**

*M. subscapularis*

This is a large, thick muscle which originates from the distal two-thirds of the subscapular fossa (Fig. 3B) distal to the presumed insertion of m. serratus ventralis. Its broad fibers are divided into four pennate portions which occupy much of the subscapular fossa. The two cranialmost pennations are broad and triangular, while the two caudalmost pennations are elongated and course parallel to the caudal border of the scapula. Its tendon contributes to the glenohumeral joint capsule and inserts onto the tip of the lesser tubercle (Fig. 5A). M. subscapularis adducts and extends the glenohumeral joint. It also stabilizes the glenohumeral joint by medially rotating the humerus in order to prevent undesired lateral rotation.

## Intrinsic muscles of the arm
### Cranial brachium

*M. biceps brachii*

M. biceps brachii has a single, spindle-shaped belly that originates from the supraglenoid tubercle of the scapula (Figs. 3A and 4). Its thick tendon crosses over the head of the humerus through the intertubercular groove of the humerus (Fig. 4), and it courses the length of the muscle. An aponeurosis covers the muscle superficially and also the proximal, deep two-thirds of the muscle (Figs. 2, 4, 6, 7 and 8). The muscle belly lies on the medial side of the humerus with fiber orientation parallel to the humerus (Figs. 4 and 7). The tendon of m. biceps brachii combines with the m. brachialis tendon before bifurcating

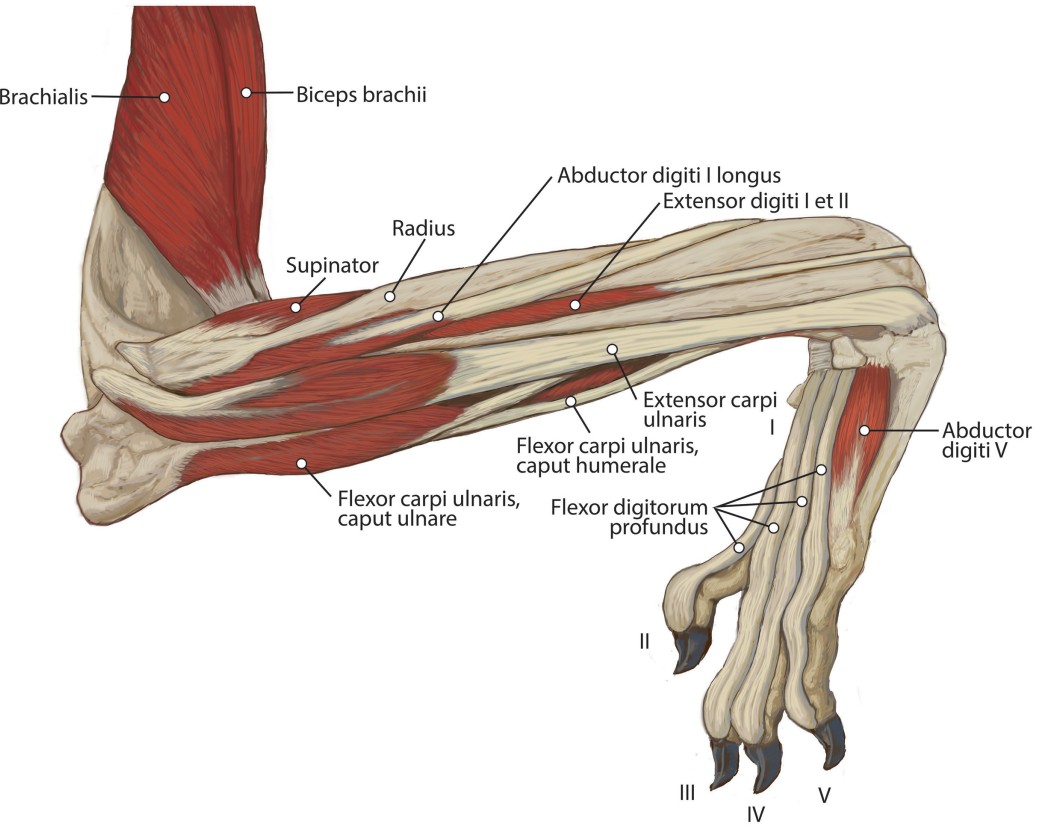

**Figure 7 Lateral view of the deep right antebrachium in *L. pictus*.**

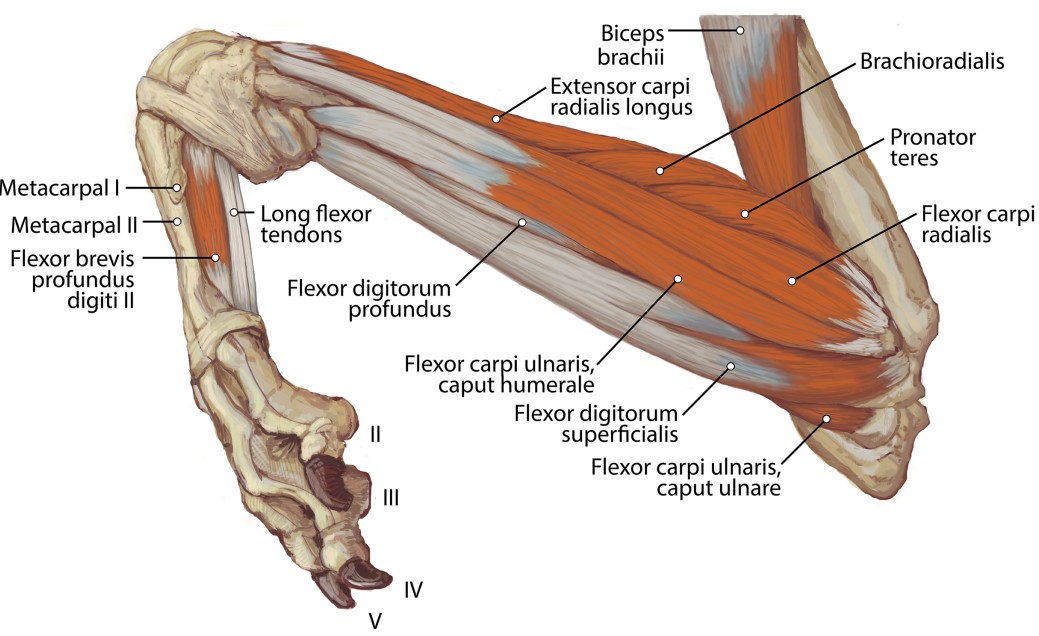

**Figure 8 Medial view of the right antebrachium in *L. pictus*, including mm. biceps brachii and brachioradialis.**

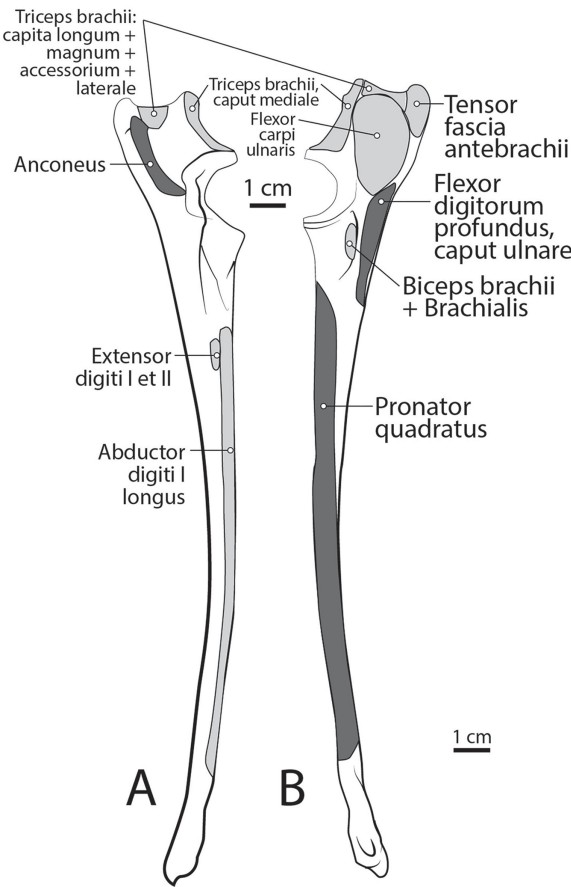

**Figure 9 Ulna muscle maps for *L. pictus* (right side): (A) lateral view; (B) medial view.**

and inserting on the radial and ulnar tuberosities (Figs. 9B and 10A). M. biceps brachii flexes the elbow and extends the glenohumeral joint. It also acts to stabilize the glenohumeral joint when the forelimb is fixed.

*M. tensor fasciae antebrachii*
This is a thin, flat, wide muscle that arises via a thin aponeurosis on the surface of m. latissimus dorsi (Fig. 4). The muscle belly tapers distally and inserts onto the medial ulna and the antebrachial fascia (Figs. 4 and 9B). This muscle tenses the antebrachial fascia and assists m. triceps brachii in extending the elbow.

*M. articularis humeri (coracobrachialis)*
This is a thin, delicate fusiform slip of muscle whose origin cannot be determined definitively due to necropsy damage (Fig. 4). The muscle belly lies over the lesser tubercle of the humerus, and is medial to the m. teres major insertion, m. triceps brachii medial head origin, and belly of m. brachioradialis (Fig. 4). It inserts onto the craniomedial aspect of the proximal humerus (Fig. 5A). M. articularis humeri stabilizes the glenohumeral joint.

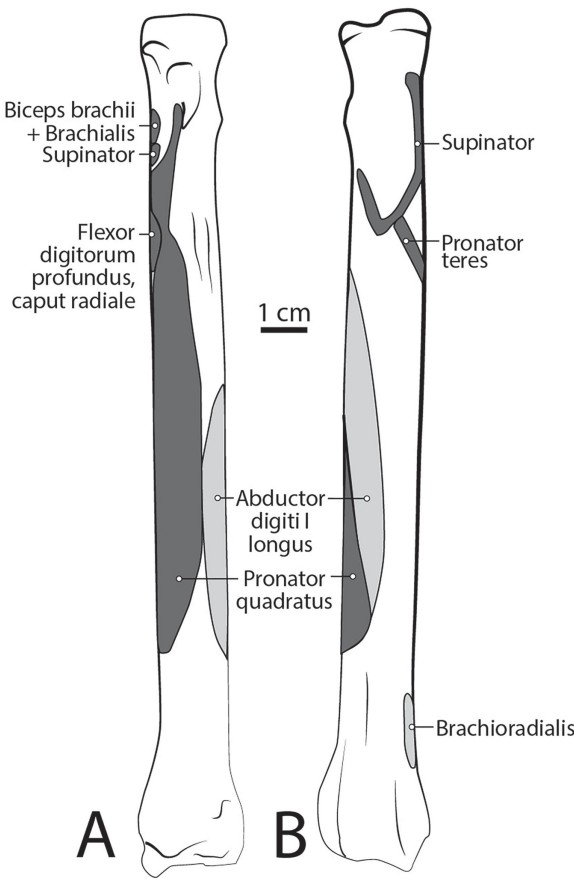

**Figure 10 Radius muscle maps for *L. pictus* (right side): (A) medial view; (B) lateral view.**

*M. brachialis*

The origin site of this muscle begins at a caudolateral divot on the proximal greater tubercle of the humerus (Fig. 5B). Its two lines of origin diverge distally from this divot. One origin follows the lateral side of the tricipital line along the deltoid tuberosity in the proximal half of the humerus and courses medially and distally to the cranial side, half-way down the humerus (Fig. 5B). The other line of origin is straight on the caudal aspect of the proximal two-thirds of the humerus, ending at the origin of the anconeus (Figs. 5A and 5B). The fibers of the muscle belly are minimally apparent, giving the belly a flat appearance (Figs. 2, 4, 6 and 7). It occupies the entire lateral brachium and passes craniomedially in its distal half (Figs. 4, 6 and 7). The muscle becomes tendinous and fuzes with the tendon of m. biceps brachii before bifurcating and inserting onto the radial and ulnar tuberosities (Figs. 9B and 10A). M. brachialis flexes the elbow joint.

**Caudal brachium**

*M. triceps brachii*

Caput magnum: It should be noted that this head is not listed in the Nomina Anatomica Veterinaria, because it was previously undescribed. We define the m. triceps brachii caput magum here as a large head of m. triceps brachii that is positioned caudally and

laterally in the m. triceps brachii complex. It originates from the caudal two-thirds of the scapular margin and neck, medial to the origin of m. teres minor (Fig. 3B). The muscle belly has a tear-drop appearance, with the tip oriented distally towards the insertion (Fig. 2). It fuses via a strong tendon with the caput longum along the distal half of the lateral aspect of the humerus (Figs. 2 and 4).

Caput longum: This head of m. triceps brachii originates from the inferior border of the scapula lateral to the origin of m. triceps brachii caput magnum (Fig. 3B). It has an elongated spindle-shaped appearance that tapers as it travels distally towards its insertion. It fuses with the caput magnum distally (Figs. 2 and 4).

Caput mediale: This head of m. triceps brachii originates from the intertubercular groove cranial to the insertion of m. teres major (Fig. 5A). The muscle is long and spindle-shaped, narrow near its origin and thicker distally (Fig. 4).

Caput laterale: The lateral head of m. triceps brachii has a thick parallelogram shape (Fig. 2). It originates via a thin, tendinous band from the proximolateral third of the humeral shaft (Figs. 2 and 9A). The muscle belly decreases in width but increases in thickness toward its insertion (Figs. 2 and 9A).

Caput accessorium: The accessory head of m. triceps brachii is long and thin with a broad origin from the lateral aspect of the humeral neck (Figs. 2 and 5B). The attachment resembles an inverted tick mark (Fig. 5B). The muscle is situated between the caput mediale and caput magnum of the m. triceps brachii (Fig. 2). Most of the muscle is on the caudal aspect of the humerus and it slightly covers m. articularis humeri and m. brachialis proximally and m. anconeus distally (Fig. 2).

The heads of m. triceps brachii fuse distally and share a stout tendon of insertion onto the entirety of the proximal portion of the olecranon process, and part of the olecranon tuber on the lateral side of the ulna (Figs. 9A and 9B). General contributions to this large insertion site include: a strong round tendon formed by the caput accessorium that enters the olecranon groove, the caput laterale insertion on the lateral side of the olecranon, and a crescent-shaped caput magnum+longum tendon that covers the tendon of the caput accessorium (Figs. 9A and 9B). Together, the m. triceps brachii are powerful extensors of the elbow and stabilize the joint during standing.

*M. anconeus*

The muscle belly of m. anconeus is triangular and generally thin (Figs. 2, 4 and 6). Its origin is V-shaped on the caudal aspect of the distal humerus and lies between the midpoint of the caudal shaft to the medial and lateral supracondylar crests (Figs. 5A and 5B). On the lateral supracondylar crest, it is adjacent to m. extensor carpi radialis and m. brachialis (Fig. 5B). There is a large fat deposit in the olecranon fossa deep to the muscle at the caudal humerus. M. anconeus has a broad, fleshy insertion from the lateral side of the olecranon to the lateral aspect of the coronoid process (Figs. 5A and 5B). M. anconeus may assist the m. triceps brachii with extension of the elbow. However, it is likely that its more important role maybe in resisting elbow flexion during standing and maintaining stability at the joint.

## Muscles of the forearm

### Caudal antebrachium

*M. brachioradialis*

This muscle has a long, thin, flat belly overlying mm. extensor carpi radialis longus and brevis (Figs. 6 and 8). It originates from approximately the lateral aspect of the distal quarter of the humerus (Figs. 5B and 6). The muscle belly is covered with an extensive fascia, and crosses from lateral to medial, distal to m. extensor carpi radialis (Fig. 6). It inserts onto the distal quarter of radius (Fig. 10B). M. brachioradialis flexes the elbow joint. It may also weakly supinate the antebrachium, but the tight attachment between the radius and ulna limits such rotatory movements.

*M. extensor carpi radialis*

This muscle is generally spindle-shaped and has a fanning origin from the lateral supracondylar ridge of the humerus (Figs. 5B and 6). An aponeurosis covers the proximal half of the caudal portion of the muscle (Figs. 6 and 8). It has two distinct bellies which are fused at the origin and separate in its distal half (Fig. 6). Each belly gives rise to a tendon in the distal third of the radius and insert as described below (Fig. 6).

*M. extensor carpi radialis longus*: The portion comprising m. extensor carpi radialis longus is approximately a third the size of m. extensor carpi radialis brevis (Figs. 6 and 8). Its muscle fibers are parallel with the radius, and it is superficial to the m. extensor carpi radialis brevis (Fig. 6). M. extensor carpi radialis longus has a smaller tendon than m. extensor carpi radialis brevis, which inserts onto the base of metacarpal II (Fig. 11).

*M. extensor carpi radialis brevis*: M. extensor carpi radialis brevis is approximately three times larger than its counterpart (Fig. 6). Its belly originates in a fan-like arrangement and gives rise to a tendon that is significantly larger than the tendon for m. extensor carpi radialis longus (Fig. 6). M. extensor carpi radialis brevis lies deep to the belly of m. extensor carpi radialis longus and has a tendinous insertion onto the base of metacarpal III (Fig. 11).

Both mm. extensor carpi radialis muscles primarily extend the carpal joint and may weakly flex the elbow.

*M. extensor digitorum communis*

This muscle originates from the proximal portion of the lateral epicondyle of the humerus (Figs. 5B and 6). A thick band of fascia covers the edges of the muscle belly proximally and also connects to m. extensor digitorum lateralis near its insertion (Fig. 6). The muscle belly is triangular and has an aponeurosis on its cranial aspect which lies deep to the m. extensor carpi radialis (Fig. 6). Its fibers converge in the middle of the muscle, and give rise to a tendon in the distal half of the radius (Fig. 6). The tendon of m. extensor digitorum communis is approximately 50% thicker than that of m. extensor digitorum lateralis (Fig. 6). The muscle remains undivided until reaching the wrist, where it divides into four tendons serving digits II–V. A dorsal sesamoid bone is embedded in each tendon as it crosses the metacarpophalangeal joint. Each sesamoid is a small, flat, elliptical bone and those of digits III and IV are largest. The tendons each insert on the extensor expansion

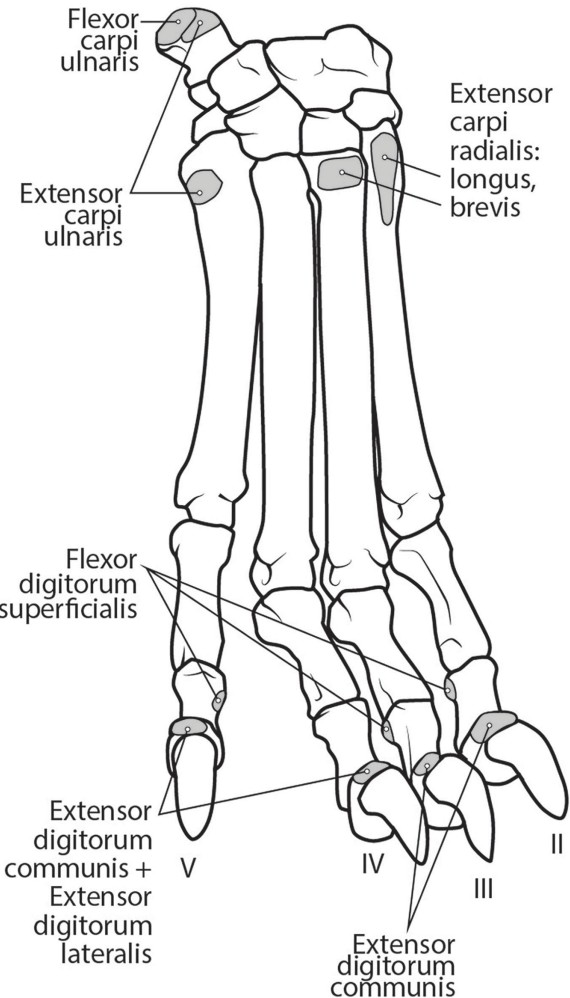

**Figure 11 Dorsal manus muscle maps for *L. pictus* (right side).**

at the distal phalanx of digits II–V (Fig. 11). The tendons of m. extensor digitorum communis attaching to metacarpal IV and V are fused with the tendons of m. extensor digitorum lateralis. M. extensor digitorum communis extends the carpal joint, metacarpophalangeal, and interphalangeal joints of digits II–V.

*M. extensor digitorum lateralis*
This muscle is fusiform and its fibers originate from the lateral epicondyle of the humerus and lateral collateral ligament (Figs. 5B and 6). Its muscle belly gives rise to a tendon in the distal half of the radius along its caudal edge (Fig. 6). The muscle is covered by an aponeurosis on its cranial and caudal edges, and it lies between m. extensor digitorum communis and m. extensor carpi ulnaris, superficial to m. extensor digiti I et II (Fig. 6). The tendons of m. extensor digitorum lateralis travel deep to m. extensor digitorum communis over metacarpal III and IV. The tendons insert on the extensor expansion at the distal phalanx of their respective digits IV–V (Fig. 11). As they do so, the two tendons fuse with m. extensor digitorum communis on the distal aspect of the proximal phalanges

of digits IV and V (Fig. 11). M. extensor digitorum lateralis does not serve digit III in either limb, which differ from the domestic dog (*Evans & De Lahunta, 2013*). This muscle extends the carpal joint, metacarpophalangeal, and interphalangeal joints of digits IV–V.

*M. extensor digiti I et II (m. extensor pollicis longus et indicis proprius)*
This muscle has an elongated, thin belly on the lateral side of the antebrachium located between the ulna and radius. It originates from the middle third of the ulna on its craniolateral aspect. The muscle lies deep to m. digitorum lateralis and travels through the concavity between the radius and ulna superficial to the interosseous membrane (Fig. 7). M. extensor digiti I et II lies immediately lateral to m. abductor digiti I longus, and its fibers travel approximately parallel to the ulna. It becomes tendinous in the distal third of the ulna (Fig. 7). Its tendon is thin and crosses medially over the dorsal aspect of the manus beginning at the wrist. The tendon bifurcates on the proximal dorsal surface of metacarpal III. One branch travels medially along metacarpal II, and at its midpoint the tendon courses deep to the tendon of m. extensor digitorum communis, before it fuses on the lateral side of m. extensor digitorum communis, on the proximal phalanx of digit II. The other belly travels further medially and also inserts on metacarpal II (Fig. 11). There is no insertion onto the vestigial digit I. M. extensor digiti I et II therefore acts to weakly extend the carpal joint, metacarpophalangeal, and interphalangeal joints of digit II, but does not act on vestigial digit I.

*M. extensor carpi ulnaris*
This is a spindle-shaped muscle originating from the lateral epicondyle of the humerus (Figs. 5B, 6 and 7). Muscle fibers are oriented caudally near the origin but become distally parallel with the ulna in its distal half (Figs. 6 and 7). The tendon arises superficially in the proximal third of the ulna but appears distally from the deep aspect in the distal quarter of the ulna (Fig. 6). The belly is covered by an aponeurosis that is thicker on its deep surface. The tendon of insertion is more robust and stiffer than those of the other antebrachial muscles. It inserts onto the lateral side of the pisiform and the base of metacarpal V (Figs. 6 and 11). In the domestic dog, the insertion on the pisiform (accessory carpal) is described as having two fiber bundles that leave the pisiform to fuse with m. extensor carpi ulnaris (*Evans & De Lahunta, 2013*). We did not observe this pattern in *L. pictus*. M. extensor carpi ulnaris abducts the carpal joint and supports the carpus, especially when it is extended and weight-bearing.

*M. abductor digiti I longus*
The muscle belly of m. abductor digiti I longus is long and flat, filling the entire interosseous space between the radius and ulna (Fig. 7). The muscle originates from the caudolateral radius, craniolateral ulna, and interosseous membrane (Figs. 9A and 10). It lies deep to m. extensor digiti I et II on the lateral side of the antebrachium in the concavity between the radius and ulna. Muscle fibers travel diagonally, oriented cranially across the forearm. A wide, flat tendon arises and tapers distally at the cranial-most aspect of the radius. The tendon crosses obliquely from lateral to medial over the tendon of m. extensor carpi radialis toward the medial side of the manus. There is a sesamoid bone in

its tendon as it becomes thinner just before its insertion onto the medial base of vestigial metacarpal I (Fig. 7). M. abductor digiti I longus primarily acts to abduct the manus. While it attaches to the vestigial metacarpal I, the bone is tightly adhered to the carpus and is unlikely to experience any notable movement at the joint between MC1 and carpals.

*M. supinator*

This deep muscle completely covers the proximal quarter of the cranial radius. It is thin, flat and fan-shaped with a heavy aponeurosis across most of its superficial surface (Fig. 7). The muscle originates from the lateral epicondyle of the humerus and lateral collateral ligament (Fig. 5B). Its fibers are oriented medially, oblique to the radius. The muscle insertion is "U"-shaped on the proximal quarter of the radius (Fig. 10B). The medial portion of the insertion extends approximately twice that of the lateral portion (Fig. 10B). Unlike the domestic dog, there is no sesamoid bone within the tendon, (*Evans & De Lahunta, 2013*). M. supinator acts to stabilize the elbow and weakly supinates the antebrachium, but limited rotatory movement is possible due to a tightly connected radius and ulna.

### Cranial antebrachium

*M. pronator teres*

The belly of m. pronator teres is spindle-shaped proximally and flatter distally (Fig. 8). An aponeurosis covers its distal half, and the muscle slightly overlaps m. supinator medially (Fig. 8). The muscle originates from the craniodistal aspect of the medial epicondyle (Figs. 5A and 8). Its fibers are obliquely oriented in a cranial direction, and it inserts onto the proximal third of the craniomedial radius, distal to the insertion of m. supinator (Fig. 10B). As its name implies, m. pronator teres weakly pronates the antebrachium. However, since antebrachial rotation is limited in *L. pictus*, its primary action may be to flex the antebrachium at the elbow.

*M. flexor carpi radialis*

This muscle is spindle-shaped. The cranial aspect of its distal half is covered by a thick aponeurosis near its origin (Fig. 8). The muscle originates from the medial epicondyle of the humerus (Fig. 5A). The caudal surface of the muscle is also covered by aponeurosis, adjacent to m. flexor digitorum profundus caput humerale (Fig. 8). The muscle becomes tendinous at the distal half of the radius (Fig. 8). The proximal muscle fibers travel parallel to the radius, but at the midline of the muscle they converge distally as they form a tendon (Fig. 8). The tendon inserts onto the palmar aspects of the bases of metacarpals II and III. M. flexor carpi radialis flexes and stabilizes the carpal joint.

*M. flexor carpi ulnaris*

This muscle includes two heads.

M. flexor carpi ulnaris caput ulnare is short and flat, originating from the distomedial aspect of the olecranon (Figs. 6 and 7). The muscle belly gives rise to a tendon at the level of the distal half of the ulna. The muscle fibers are oriented slightly obliquely toward the medial side. M. flexor carpi ulnaris caput ulnare caput ulnare lies superficial to m. flexor

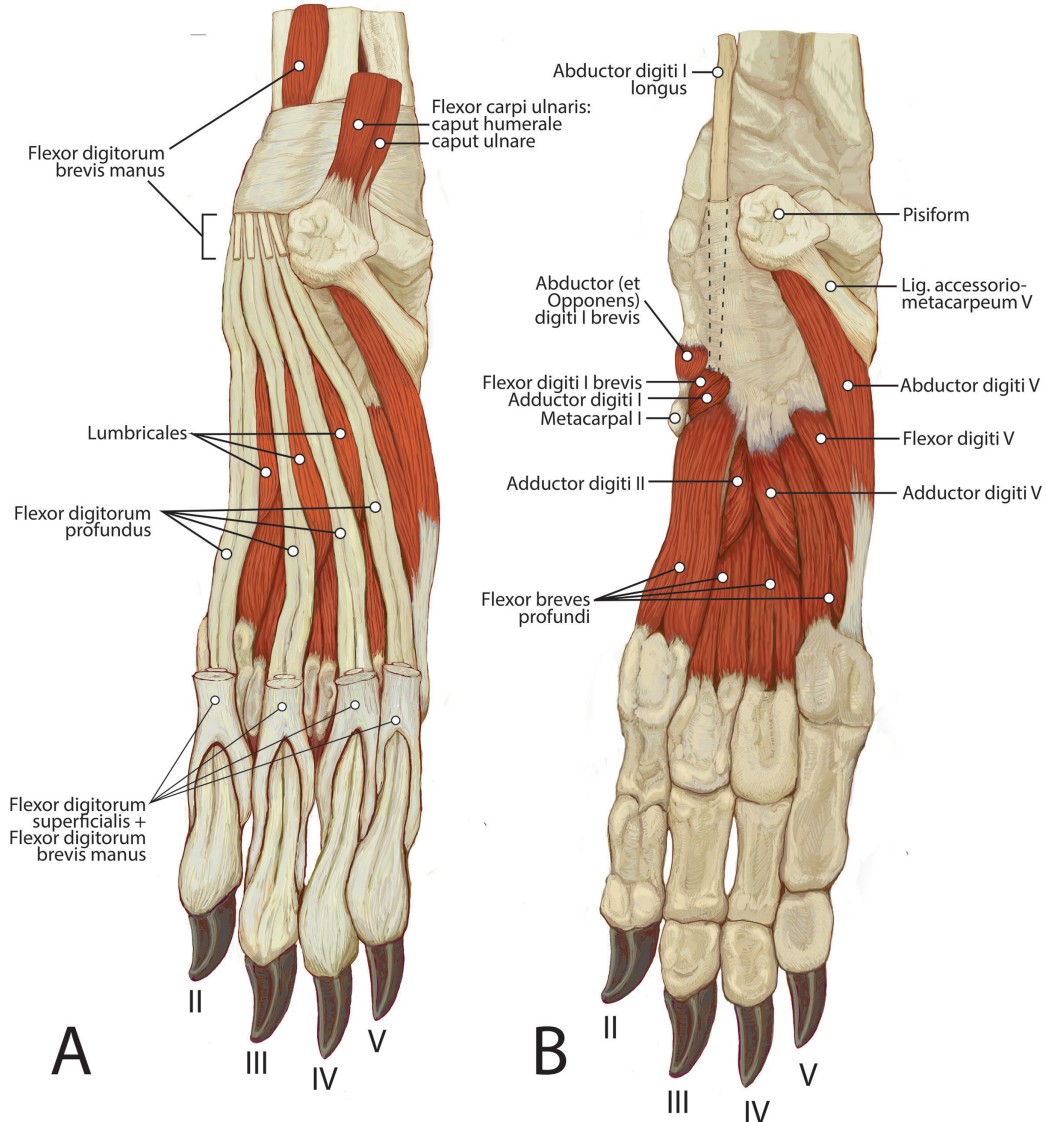

**Figure 12 Palmar view of manus muscles in *L. pictus* (right side): (A) superficial view; (B) deep view.**

digitorum profundus caput ulnare and is lateral to m. flexor digitorum superficialis (Figs. 6 and 7). It inserts on the pisiform (accessory carpal bone), cranial to the insertion of m. flexor carpi ulnaris caput humerale (Figs. 11 and 12).

M. flexor carpi ulnaris caput humerale is flat proximally, becoming spindle-shaped in its distal half (Figs. 6 and 7). It originates from the caudal aspect of the medial epicondyle of the humerus (Figs. 4 and 5). Fascia covers most of the muscle on its superficial and deep surfaces (Figs. 6 and 7). The muscle fibers are oriented parallel to the ulna, and the muscle inserts onto the pisiform, cranial to the insertion of m. flexor carpi ulnaris caput humerale (Figs. 11 and 12).

Together the two heads of m. flexor carpi ulnaris flex and abduct the manus.

*M. flexor digitorum superficialis*

This muscle is spindle-shaped and somewhat flat (Figs. 8 and 12). It is superficial to m. flexor digitorum profundus and m. flexor carpi ulnaris caput humerale (Fig. 8). The muscle originates from the medial epicondyle of the humerus (Fig. 5). The muscle becomes tendinous at the wrist and travels deep to the flexor retinaculum before passing medial to the pisiform (Figs. 11 and 12). It then bifurcates over the proximal phalanges of digits II–V, allowing passage of m. flexor digitorum profundus. The muscle inserts bilaterally on the palmar aspects of the middle phalanges of digits II–V (Figs. 11 and 12). M. flexor digitorum superficialis flexes the carpal, metacarpophalangeal, and proximal interphalangeal joints of digits II–V.

*M. flexor digitorum profundus*

M. flexor digitorum profundus has five heads of various sizes, including three with a shared humeral origin, one ulnar, and one radial. It serves digits II–V and does not contact the vestigial digit I (Figs. 5, 8, 9, 10 and 12).

Caput humerale laterale: This is the largest of the humeral heads, and accounts for approximately half of the entire humeral component, which has a fused origin (Fig. 8). It is spindle-shaped, with fibers that fan out from its origin at the caudodistal aspect of the medial epicondyle (Fig. 5). It lies adjacent to the caput humerale mediale, to the caput profundus, and deep to m. flexor digitorum superficialis (Fig. 8). Approximately three-quarters of the medial surface of the muscle is covered by parallel-fibered aponeurosis.

Caput humerale mediale: This is the most superficial of the humeral heads (Figs. 5 and 8). It is spindle-shaped, covered by a thick aponeurosis, and about half the size of the caput humerale laterale. This medial head lies lateral to m. flexor carpi radialis, medial to caput humerale laterale and m. flexor digitorum superficialis, and superficial to caput humerale profundus.

Caput humerale profundus: This is the smallest humeral head. It is spindle-shaped and its fibers travel parallel to the ulna (Fig. 5). The medial aspect of its proximal half is covered by an aponeurosis. It gives rise to a tendon along the distal third of the radius, which is enclosed by the capita humerale laterale and mediale.

Caput ulnare: This belly lies caudal to the ulna (Fig. 9B). It originates from the base of the olecranon and the proximal quarter of the ulnar shaft (Fig. 9B). The entire muscle belly is slightly adhered via fascia to the ulna. Its tendon arises unattached to the bone, and joins the three humeral heads. The muscle's fibers are parallel to the ulna, and a thick aponeurosis covers the medial half of the superficial surface.

Caput radiale: This is the smallest head of m. flexor digitorum profundus, located on the medial side of the antebrachium, superficial to m. pronator quadratus. Its thin muscle belly originates from the proximal third of the caudomedial radius (Fig. 10A). The muscle fibers travel parallel to the radius, and the insertion tendon arises at the level of the distal quarter of the bone.

Tendons of the flexor digitorum profundus fuse at the distalmost radius, and the combined tendon passes deep to the flexor retinaculum and medial to the pisiform. It divides into four smaller parts deep to the retinaculum, which serve digits II–V

(Figs. 10 and 11). The tendon of caput ulnare primarily serves digit V, but such a distinction cannot be made for the tendons serving digits II–IV. M. flexor digitorum profundus flexes the carpal, metacarpophalangeal, proximal interphalangeal, and distal interphalangeal joints of digits II–V.

*M. pronator quadratus*
This muscle lies deep to m. flexor digitorum profundus, occupying the entire space between the radius and ulna on the medial side of the antebrachium. It originates from the craniomedial aspect of the distal ulna and interosseous membrane (Fig. 9B). Its fibers are oriented obliquely to perpendicular to the radius and ulna. An aponeurosis covers the superficial surface of the proximal three quarters of the muscle belly. The muscle inserts on almost the entire caudomedial aspect of the radial shaft (Fig. 10). Unlike the muscle attachments in the domestic dog, these are situated more proximally on the radius and ulna (*Evans & De Lahunta, 2013*). M. pronator quadratus weakly pronates, but likely primarily acts to stabilize the antebrachium.

## Muscles of the manus

*M. palmaris brevis*
This muscle was not observed in *L. pictus*.

*M. flexor digitorum brevis manus (m. interflexorius)*
This muscle originates from a fascial sheet from m. flexor digitorum profundus capita humerale mediale and laterale in the distal one-quarter of the antebrachium (Fig. 12A). It lies on the palmar side of m. flexor digitorum profundus, deep to m. flexor digitorum superficialis, and it courses with the digital flexors through the carpus deep to the flexor retinaculum (Fig. 12A). The muscle belly is fusiform, becoming tendinous immediately proximal to the pisiform. the tendon splits and fuses onto the distal tendons of m. flexor digitorum superficialis serving digits II–V inserting with the latter (Fig. 12A). This muscle therefore assists m. flexor digitorum superficialis in flexing the carpal, metacarpophalangeal, and proximal interphalangeal joints of digits II–V.

*Mm. lumbricales*
These three thin muscles originate from the tendons of m. flexor digitorum profundus. They are superficial to mm. flexores breves profundi (Fig. 12A). The muscle fibers course parallel to the long axes of the metacarpals between adjacent tendons of m. flexor digitorum profundus (Fig. 12A). The m. lumbricalis belly between the m. flexor digitorum profundus tendons to digits III and IV is the largest, while the bellies between the tendons of digit II and III and between IV and V are smaller (Fig. 12A). The tendons of mm. lumbricales wrap dorsally to insert onto the medial sides of the extensor expansions of digits III-V (Fig. 12A). The mm. lumbricales flex the metacarpophalangeal joints and extend the interphalangeal joints of digits III–V.

*Mm. adductores digitorum*
The three mm. adductores digitorum have adjacent origins from the ligamentum carpi transversum before traveling to digits I, II and V (Fig. 12B).

*M. adductor digiti I (m. adductor pollicis)*

This small belly originates from the ligamentum carpi transversum cranial to m. flexor brevis profundus to digit II (Fig. 12B). It is a short, fan-shaped muscle that is wide at the origin and tapers at its insertion (Fig. 12B). It is superficial to m. flexor brevis profundus I and lateral to m. abductor digiti I brevis and m. flexor digiti I brevis (Fig. 12B). It inserts onto the palmomedial surface of the vestigial first metacarpal distal to the insertion of m. abductor (et opponens) digiti I (Fig. 12B). As there appears to be little mobility of the vestigial MC1, m. adductor digiti I may stabilize the reduced digit I and carpus.

*M. adductor digiti II*

This belly originates from the ligamentum carpi transversum between m. flexor brevis profundus digiti I and m. adductor digiti V (Fig. 12B). It is a relatively thick band of muscle lying on the palmar side of the carpometacarpal joint (Fig. 12B). Its muscle fibers are parallel with the metacarpals (Fig. 12B). The muscle lies deep to m. flexores breves profundi II and III and courses distally between metacarpals II and III (Fig. 12B). The muscle becomes tendinous at the distal aspect of the metacarpophalangeal joint and inserts onto the palmolateral base of the proximal phalanx of digit II (Fig. 12B). M. adductor digiti II adducts digiti II and weakly flexes its metacarpophalangeal and proximal interphalangeal joint.

*M. adductor digiti V*

This muscle originates from the pisiform and center of the ligamentum carpi transversum on the palmar aspect of the carpometacarpal joint (Fig. 12B). It is a flat, fan-shaped muscle with slight aponeurosis on the proximal two-thirds of its palmar surface (Fig. 10B). It inserts onto the distomedial aspect of metacarpal V and partially attaches to the base of the proximal phalanx of digit V (Fig. 12B). M. adductor digiti V adducts digit V and weakly flexes the associated metacarpophalangeal and proximal interphalangeal joints.

*Mm. flexores breves profundi (mm. interossei)*

The four mm. flexores breves profundi originate from the bases of metacarpals II–V and also from the carpometacarpal joint capsules of digits II–V (Fig. 12B). They are located palmar to metacarpals II–V and lie deep to mm. adductores digitorum, the tendons of mm. flexor digitorum profundus and superficialis (Fig. 10B). At their distal third, each m. flexor brevis produndus splits into medial and lateral bellies that diverge to either side of the associated metacarpal (Fig. 12B). At the metacarpophalangeal joint, each belly abruptly becomes tendinous and inserts onto the medial and lateral proximal sesamoids (Fig. 12B). At the middle of metacarpal II, the tendons wrap collaterally to the dorsal side of the manus and fuse with the tendons of m. extensor digitorum communis (Fig. 12B). This pattern was not observed in any other digit but may perhaps vary among individuals. The mm. flexores breves profundi flex the metacarpophalangeal joints of digits II–V. In those digits in which the tendons also insert onto the extensor expansion, they may also extend the interphalangeal joints.

*M. flexor digiti I brevis*

This muscle originates from the flexor retinaculum proximal to the origin of m. flexor brevis profundus to digit II (Fig. 12B). It is a short, thick muscle with muscle fibers traveling parallel to the metacarpals (Fig. 12B). Distally, it dives deep to the vestigial metacarpal I (Fig. 12B). Since minimal mobility is possible at the vestigial MC1, m. flexor digiti I brevis may stabilize the reduced digit I and carpus.

### M. abductor (et opponens) digiti I

This short, cylindrical muscle originates from the flexor retinaculum (Fig. 12B). It lies superficial to m. flexor digiti I brevis and inserts onto fascia superficial to vestigial metacarpal I (Fig. 12B). Unlike in the domestic dog, there is no osseous insertion. Due to its lack of bony attachment, this muscle may weakly stabilize the wrist.

### M. abductor digiti V

This is a small, flat muscle that arises from the distal aspect of the pisiform (Fig. 12B). Its fibers course obliquely and laterally from the pisiform to the lateral manus (Fig. 12B). The muscle produces a thin, transparent aponeurosis at the proximal quarter of metacarpal V, which courses distally and inserts on the lateral base of the proximal phalanx of digit V (Fig. 12B). M. abductor digiti V abducts digit V and flexes the metacarpophalangeal joint.

### M. flexor digiti V

This muscle originates from the flexor retinaculum of the pisiform. It is a small muscle that lies distal to the origin of m. abductor digiti V and travels laterally over m. flexor digitorum brevis digiti IV (Fig. 12B). The muscle becomes tendinous and fuses with the tendon of m. abductor digiti V, then inserts onto the base of the proximal phalanx of digit V (Fig. 12B). M. flexor digiti V flexes and abducts digit V.

## Noteworthy ligaments

The ligaments of the upper limb of *L. pictus* were generally comparable to those of other published canids and are thus presented in detail in the Supplemental Information (Table S1). However, three notable ligamentous differences are worthy of full consideration here.

### Membrana interossea antebrachii

This is a thick syndesmosis between the diaphyses of the radius and ulna. It courses along the entirety of both bones from immediately distal to the radial head through to the radiocarpal joint. The ligament holds the radius and ulna in a tight connection such that very little pronation or supination is possible. The membrane contained no apparent perforations for neurovasculature. The m. pronator quadratus attaches along almost the entire length of the membrane.

### Ligamentum interossei antebrachii

This short, thick ligament tightly connects the radius and ulna. At approximately one-third of the way along the antebrachium, it courses from a roughened area on the lateral

radial shaft to a roughened area on the ulnar shaft. Its fibers run slightly obliquely distally. It is approximately 3.5 cm in length, substantially longer than the 2 cm length of the domestic dog. On the right side, it was partially ossified at its attachment to the ulna. As with the membrana interossea antebrachii, this ligament prevents rotatory movement of the radius around the ulna.

### Ligamentum accessoriometacarpeum (pisimetacarpeum) V

This ligament is incredibly robust, 5.3 mm thick and 18.5 mm long. It attaches proximally to the lateral surface of the pisiform (accessory carpal bone). Its fibers course parallel to the orientation of the antebrachium and attach distally to the lateral aspect of the base of metacarpal V (Fig. 11). Unlike the domestic dog, there is no attachment to metacarpal IV. This is the thickest and most powerful of the carpal ligaments. The natural position of the carpal joint in *L. pictus* appears to be in a flexed position, held tightly by the ligamenta accessoriometacarpea. The pisiform is more heavily involved in the ligamentous apparatus of the carpus than in domestic dogs.

## DISCUSSION

### Vestigial digit I

An unexpected finding of the present study was the presence of a small vestigial first metacarpal in *L. pictus*, challenging previous assumptions of complete tetradactyly in this species. While metacarpal I is diminutive and not associated with any phalanges, it still serves as the attachment site for several pollical muscles. Thus, we interpret *L. pictus* as being functionally, but not fully morphologically, tetradactyl.

The absence of a first digit in *L. pictus* has been argued to facilitate increased speed and stride length (*Creel & Creel, 2002*; *Chavez et al., 2019*). The vestigial digit I of *L. pictus* lacks a claw and thus cannot grip the substrate during locomotion; however, the muscles associated with digit I may provide additional stability to the forefoot during endurance running. While early studies on energy expenditure estimated that energy costs for *L. pictus* during hunting are high (*Gorman et al., 1998*), recent studies based on more comprehensive models have concluded that hunting is less energetically costly for *L. pictus* than originally calculated (*Hubel et al., 2016*). Although *L. pictus* groups may travel an average of 14 km per day, including long-distance endurance chases (*Hubel et al., 2016*), they also utilize short high-speed bursts of opportunistic predation. These energetically efficient hunting opportunities effectively balance out the energy expended during exhaustive predation episodes. In another digitally reduced mammalian lineage, Equidae, the vestigial second and fourth metacarpals serve an important function in supporting the articulation of the carpus and manus (*Janis & Bernor, 2019*). Another possible function for the reduced pollical muscles in *L. pictus* is proprioception. The diminutive size of the muscle bellies suggests that they provide minimal mechanical leverage. However, muscles such as m. abductor digiti I with its long thin tendon, may assist with proprioceptive functions around the carpal joint, which could be significantly advantageous during exhaustive predation.

## Quantitative forelimb muscle proportions

The quantitative muscle proportion analyses indicated that canids generally possess relatively smaller pronators and supinators than felids, which is unsurprising given their divergent uses of the forelimb in locomotion and prey capture and manipulation. However, *L. pictus* is notable, even among canids, in its diminutive rotatory muscle mass. *L. pictus* has a smaller proportion of wrist rotators than any other comparative carnivoran taxon, followed by other large-bodied terrestrial taxa known to run and travel long distances. This finding supports the anatomical descriptive analyses which revealed a radius and ulna that were tightly adhered by a robust interosseous membrane and interosseous ligament. Minimal rotatory movement is possible between the radius and ulna in *L. pictus* suggesting adaptations for stability in the forelimb.

Additionally, the muscles associated with digit I were found to be reduced in *L. pictus* compared to all other taxa in the sample, exaggerating the reduced condition observed in other canids. In particular, mm. abductor digiti I and extensor digiti I et II are smaller and their attachments sites differ due to the absence of a full digit I. M. extensor digiti I et II does not insert onto the vestigial first digit, and instead possesses only a single tendon inserting onto digit II. While the belly of m. abductor digiti I is broad, its tendon is gracile and thin, and has only a small attachment site onto fascia overlying the base of the vestigial metacarpal I.

The similarities in forelimb muscle proportions between *L. pictus* and the dhole (*Cuon alpinus*) provide further insight into adaptations to cursoriality in canids. While not as highly cursorial as *L. pictus*, *C. alpinus* is also hypercarnivorous, hunts communally, and focuses on medium to large-bodied ungulates (*Durbin et al., 2004*). Most of its predatory chases are short, around 500 m (*Fox, 1984*), but it has been documented occasionally chasing prey for hours (*Heptner & Naumov, 1998*). However, *C. alpinus* has a fully formed digit I (*Castello, 2018*) making it an interesting comparative taxon. In both *L. pictus* and *C. alpinus*, m. flexor digitorum profundus, m. extensor digiti I et II, and abductor digiti I longus are reduced compared to all other included taxa, likely reflecting the reduced need for fine digital manipulation in these cursorial taxa. They also share relatively smaller mm. supinator and pronator teres, further supporting the interpretation that reduction of the rotatory muscles facilitates stability in the forelimb of long-distance runners. However, it should be noted that these reductions are more pronounced in *L. pictus*. The two species share an enlarged m. supraspinatus, which is an important stabilizer of the glenohumeral joint and acts to prevent collapse of the shoulder (*Goslow et al., 1981*).

## Comparisons of qualitative muscular morphology to other published carnivoran taxa

### Musculature of the vestigial digit I

The forelimb myology of *L. pictus* is generally similar to other canids; however, it displays some notable exceptions. The presence of a vestigial first metacarpal results in changes to insertions of mm. extensor digiti I et II, adductor digiti I longus, abductor (et opponens) digiti I, and flexor digiti I brevis compared to other canids. The origin of m. extensor digiti I

et II is smaller than in the domestic dog, *Canis familiaris* (*Evans & De Lahunta, 2013*) and pampas fox (*De Souza Junior et al., 2018*), and it inserts exclusively onto digit II, bypassing the vestigial digit I entirely. Interestingly, the tendon of insertion still bifurcates, but rather than inserting onto digit I as in other canids, one tendon fuses with the insertion of extensor digitorum communis to digit II and the other inserts directly onto metatarsal II. Thus, the muscle does not act on the vestigial first digit. M. adductor digiti I longus of *L. pictus* has a substantially reduced origin compared to other canids, and its tendon of insertion onto metacarpal I is wispy and diminutive. It appears unlikely that it has sufficient leverage to move MC1.

## Muscles and ligaments involved in forelimb stability and endurance running

The m. triceps brachii complex is expanded in *L. pictus* compared to other canids. In particular, m. triceps brachii caput laterale has a larger origin in *L. pictus* compared to other carnivorans, and m. triceps brachii long head consist of two large bellies, referred to herein as caput longum and caput magnum. Electromyographic (EMG) studies have shown that m. triceps brachii, especially caput laterale, is active during the stance phase of trotting and galloping and is important for storing elastic energy during locomotion (*Goslow et al., 1981*). Around the elbow, m. anconeus also has a more extensive and proximal origin in *L. pictus* compared to other canids. In addition to extending the elbow, this muscle plays an important role in resisting elbow flexion. It is composed almost entirely of Type I fibers and has been suggested to provide proprioceptive information about the elbow joint to the central nervous system (*Buxton & Peck, 1990*).

M. brachialis has a more extensive origin in *L. pictus* than in other canids. In *C. familiaris*, this muscle has relatively high proportion of Type I "slow-twitch" fibers but is also electrically active during the swing phase of locomotion (*Goslow et al., 1981*). M. flexor carpi ulnaris caput ulnare of *L. pictus* is a larger muscle with a more extensive origin than *C. familiaris*. In domestic dogs, this muscle belly contains a large percentage (mean 77%) of Type I muscle fibers, suggesting that it is resistant to fatigue (*Armstrong et al., 1982*). Also, in addition to its role in flexion of the forepaw with abduction, it has also been argued to play an important postural role in the forelimb (*Evans & De Lahunta, 2013*). While EMG studies and assessments of Type I vs. Type II fibers in the forelimb muscles of *L. pictus* are out of the scope of the present study, it is reasonable to assume that the general patterns would not be dramatically divergent from those of *C. familiaris*. M. flexor digitorum profundus capita ulnare and radiale have smaller origins in *L. pictus* than all other comparative taxa, and the muscle's combined bellies have a smaller mass than most other comparative carnivoran taxa. This may reflect a reduced need for fine digital flexion. The mm. biceps brachii and brachialis appear to function as a unit, fusing distally to insert together onto both the radius and ulna. This pattern contrasts with the domestic dog in which the m. brachialis tendon courses through a split in that of the m. biceps brachii to insert exclusively onto the ulna (Fig. 13). This difference may provide additional stability at the elbow joint in *L. pictus*. Overall, the forelimb muscles of *L. pictus* demonstrate a definitive pattern of stability and resistance to fatigue, with a concomitant reduction of

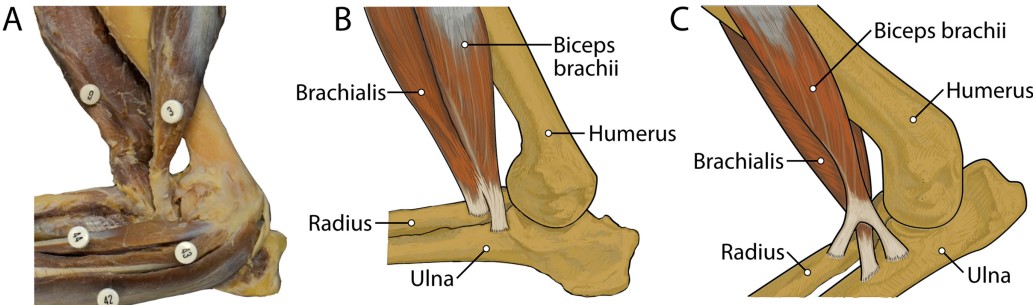

**Figure 13 Insertions of mm. biceps brachii and brachialis: (A) photograph of *Lycaon pictus*, (B) illustration of *L. pictus*, (C) illustration of *Canis familiaris*.** Numbers indicate: 3, m. biceps brachii; 9, m. brachialis; 42, m. flexor digitorum superficialis; 43, m. flexor digitorum profundus; 44, m. flexor carpi radialis. Note that in *L. pictus*, the tendons of mm. biceps brachii and brachialis fuse to insert together onto both the radius and ulna, while in *C. familiaris* the m. brachialis tendon travels through a split in the tendon of m. biceps brachii to insert exclusively onto the ulna.

rotatory movements, mobility of digit I, and fine digital flexion. The membrana interossea antebrachii and ligamentum interossei antebrachii are expanded compared to other canids, adhering the radius and ulna tightly together.

The extensor and flexor muscles of the carpus in *L. pictus* are generally comparable in size and attachments to other canids. However, the wrist is supported by an incredibly stout ligamentum accessoriometacarpeum V attached to a prominently projecting pisiform, which may act as a strut for assisting with passive flexion and rebound of the forefoot during sustained locomotion. During touchdown of the manus, the carpus becomes extended passively due to gravity. The natural tautness of the robust ligamentum accessoriometacarpeum V would tend to pull the wrist back into flexion as soon as the forefoot starts leaving the ground, likely providing non-muscular propulsion during push-off. This passive mechanism may help sustain endurance running and prevent the wrist muscles from tiring. This morphology displays apparent functional convergence with the suspensory ligaments of the equid "spring foot". The equid suspensory ligament is derived from the third interosseous muscle, and functions to support the forefoot and provide passive "spring" action by absorbing and transferring forces experienced during locomotion (*McGuigan & Wilson, 2003*). In the similarly cursorial *L. pictus*, the enlarged ligamentum accessoriometacarpeum V may serve a comparable function.

We interpret these differences in size and attachments of muscles in *L. pictus* compared to other canids as adaptations for long distance running in this highly cursorial species, likely important for exhaustive predation. Absence of a complete digit I in *L pictus*, typically used to reduce torque during quick turns and for lightly gripping onto objects, may be related to a reduced need for gripping and quick agile movements in its cursorial lifestyle.

## Limitations

Finally, it must be acknowledged that this study is based on two limbs from a single captive specimen. Although this scenario is not ideal, specimens of this endangered species are

extremely difficult to obtain. Thus, inferences must be drawn from a single individual that is assumed to be representative of the species. Of particular consideration are attachments sites of muscles which have been found to vary in other canids. For example, attachments of several palmar muscles display intraspecific variation in domestic dogs and Pampas fox (*De Souza Junior et al., 2018*). However, while we fully recognize the inability of the present study to evaluate intraspecific variation, the information presented herein may still inform future comparative studies of carnivoran adaptation and evolution. Future studies including additional *L. pictus* specimens may contribute to the understanding of variation in this taxon.

## CONCLUSIONS

A vestigial first digit was discovered in *L. pictus*, in the form of a diminutive first metacarpal, demonstrating for the first time that this species is not fully tetradactyl. The forepaw is supported by a stout ligamentum accessoriometacarpeum V, which holds the wrist in passive flexion. Wrist rotator muscles are reduced compared to other carnivorans, and robust interosseous ligaments bind the radius tightly to the ulna. Several muscles associated with joint stability and known to store elastic energy are expanded. The unexpected metacarpal I results in dramatic morphological alterations to associated digit I musculature. Natural tautness of ligamentum accessoriometacarpeum V may provide passive propulsion during the toe-off phase of locomotion, helping to sustain endurance running. These traits represent adaptations for long distance running and would facilitate exhaustive predation.

## ACKNOWLEDGEMENTS

The *L. pictus* specimen in this study was donated to the ARCIVES by the Arizona Center for Nature Conservation/Phoenix Zoo. The authors wish to thank M. Taverne for generously sharing muscle mass data for *Cuon alpinus* and *Vulpes vulpes* for the comparative analyses. We are grateful to Drs. Suvi Viranta, Damian Ramoni, and Brandon Hedrick whose insightful comments improved the manuscript. HFS would like to thank Dr. Kaye Reed for the introduction to this amazing species in Kruger National Park in the summer of 2000. This is ARCIVES Publication #4.

### Funding

This study was funded by a Midwestern University Kenneth A. Suarez Summer Research Fellowship to Rahul Koshy. The funders had no role in study design, data collection and analysis, decision to publish, or preparation of the manuscript.

### Grant Disclosures

The following grant information was disclosed by the authors:
Midwestern University.

## Competing Interests

The authors declare that they have no competing interests.

## Author Contributions

- Heather F. Smith conceived and designed the experiments, performed the experiments, analyzed the data, prepared figures and/or tables, authored or reviewed drafts of the paper, and approved the final draft.
- Brent Adrian conceived and designed the experiments, analyzed the data, prepared figures and/or tables, authored or reviewed drafts of the paper, and approved the final draft.
- Rahul Koshy performed the experiments, authored or reviewed drafts of the paper, and approved the final draft.
- Ryan Alwiel performed the experiments, authored or reviewed drafts of the paper, and approved the final draft.
- Aryeh Grossman analyzed the data, authored or reviewed drafts of the paper, and approved the final draft.

## Data Availability

The raw data are available in Tables 1–3 and Table S1.

The full STL and OBJ files of the CT scans are available at MorphoSource:
https://www.morphosource.org/Detail/SpecimenDetail/Show/specimen_id/32229.

—M66470-119450: http://www.morphosource.org/Detail/MediaDetail/Show/media_id/66470.

—M66470-119455: http://www.morphosource.org/Detail/MediaDetail/Show/media_id/66470.

## Supplemental Information

Supplemental information for this article can be found online at http://dx.doi.org/10.7717/peerj.9866#supplemental-information.

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
