# Peer review of "Adaptations to cursoriality and digit reduction in the forelimb of the African wild dog (Lycaon pictus)"

_PeerJ, doi:10.7717/peerj.9866_

## Round 0.1 · original submission · Major Revisions

I commend the authors on a detailed analysis of the African wild dog musculature and agree with the authors that this work will be valuable to conservation scientists and exotic vets in addition to evolutionary biologists. Based on two reviews, I feel that the article requires several moderate revisions before it can be published.

In particular, the authors must be more up front about how this analysis is based on a single individual. I would suggest adding a paragraph on this point to the discussion. I would also encourage the authors to remove the reference to Zink (2005). I found this article online, but it does not appear to be peer reviewed. This will require some reworking of the introduction (it can be supplanted by the Chavez et al. (2019) paper that is mentioned by reviewer 2 and the movement of the fossil discussion from the discussion to the introduction as suggested by reviewer 1).

In addition to the points discussed by the reviewers, it would be good to have more detail on muscle mass measurements, particularly since the forelimbs were preserved in formalin for a period of time. This can lead to some shrinkage, which should be discussed (Fox et al., 1985). I don’t think that this would have had much of an impact, but it needs to be addressed.

Fox CH, Johnson FB, Whiting J and Roller PP (1985) Formaldehyde fixation. J Histochem Cytochem 33, 845–853.
Here are a few minor points that I caught as well:

Line 138: comparative should not be capitalized

Line 168: regression analysis should not be capitalized. Was this an OLS regression? More detail here is needed.

Should be principal component analysis rather than ‘components’

I look forward to seeing your revision!

·

Basic reporting

No comments

Experimental design

No comments

Validity of the findings

No comments

Additional comments

The article submitted by Smith and collaborators about the anatomy and morpho-functionality of Lycaon's leg is a relevant and useful article that I think should be published. However, there are some suggestions.
In general, the “Fossil record of Lycaon spp.” section is not relevant to the article. Only the sentence “The vestigial metacarpal I of L. pictus, discovered here, is approximately 30% the length of its metacarpal II, compared to 38% in L. sekowei (Fig. 14). This pattern indicates a functional loss, but not complete loss, of metacarpal I over time in the Lycaon lineage”, points out aspects that are based on the revised material. And about this sentence, must be taken delicately because it is only based on one measure of a single current specimen. This is not necessarily true because the article does not consider potential intra-specific variability.
I suggest migrating the section of the fossil record to the introduction if authors feel that they can use bone morphology to validate their form-functional conclusions. The article focuses on the description of the muscles and little on the description of the bones.
Finally, in some cases, the use of taxonomic nomenclature is misused. Eg., italics where it should not be italicized.

·

Basic reporting

A recent publication by Chavez et al. (2019) provides a genetic and developmental basis for the rudimentary (not lost!) MCI. This is very much in line with the ms, but authors should discuss these data in the light of their results. Currently it is not recognized in the paper.

The authors provide possible function for the first digit (dew claw) both in the introduction and discussion referring to Zink 2005. This reference is impossible to locate using standard literature engines or databases, and may not be scientifically reliable. Use of the dew claw in high speed turns as a grip is questionable.

The caput magnum m. triceps brachii needs to be defined. While it is ok to use a name not listed in Nomina Anatomica Veterinaria, it is advisable to explain the name and its use.

The ancestry of Cuon from Xenocyon is not supported by presence of vestigial MCI in latter, as the former has fully developed MCI. Please remove or rewrite.

There are some minor problems and mistakes in the figures that need to be addressed:
Fig 2. is a medial view
Fig 4 is a lateral view and deltoideus pars scapularis appears to be not right.
Fig. 5. for m. pectoralis superficialis use the same color in cranial and caudal views.
Fig 8. the long flexor tendon does not look right. Please redraw or relabel.

Ref:
Chavez, D. E., Gronau, I., Hains, T., Kliver, S., Koepfli, K. P., & Wayne, R. K. (2019). Comparative genomics provides new insights into the remarkable adaptations of the African wild dog (Lycaon pictus). Scientific reports, 9(1), 1-14.

Experimental design

The main weakness of the data is the sample size: one individual, two limbs.
While this is common and understandable when working on endangered species and does not prevent from publishing the data, the authors should make it more clear that there most likely is variation in the structures that were not revealed here. Moreover, the specimens are from a captive individual which may have slight influence on e.g. muscle masses, and this should be brought up. Also the body condition of the animal could be presented in the materials.

Validity of the findings

no comment

Additional comments

This is a very nice piece of work, and it is good to get these data and the analysis out for the African wild dog. The main problem of the paper is the specimen number which is limited to one individual. Please emphasize this uncertainty more in the discussion and interpretation. The muscle (especially the palmar!) insertion sites and numbers are known to show intraspecific variation in the domestic dog and this was also the case e.g. in the Pampas fox (Souza Junior et al 2018).

You suggest the unique strong ligaments and tendons to function as energy saving springs during fast locomotion, and this is a valid interpretation. The other aspect, the enhanced need for proprioception, is not discussed. Structures like the adductor digit 1 with its thin tendon could serve mainly a proprioceptive purpose. Please consider this when revising your manuscript.
It is tempting to compare these tendons to the equid suspensory ligament and its evolution from the mm interossei. The ligament in equids is important not only in energy saving but also for safety by providing proprioceptive feedback.

---

## Round 0.2 · accepted · Accept

Thanks so much for your submission. I am happy to move this paper forward. It is an interesting and important contribution to the literature. There are a few minor points that you should correct before the paper is published, but I have listed this as 'accept'.

Line 18: Canidae and Caniformia should be capitalized in the key words

Line 232: should be Taverne et al. (2018) rather than Taverne and colleagues

Line 433: magnum spelling

Line 797: Period at the beginning of the sentence

Line 808: Equidae should be capitalized

Line 918: ‘accesoriometacarpeum’ should be ‘accesoriometacarpal’

Line 936: Space between limitations paragraph and conclusions

From Reviewer 2:
Line 490 'MC1-carpojoint' does not sound right, I think it should be either metacarpocarpal joint or joint between MC1 and carpals.

·

Basic reporting

No comment

Experimental design

No comment

Validity of the findings

No comment